# NoisePrints: Distortion-Free Watermarks for Authorship in Private Diffusion Models

**Nir Goren**[1] **Oren Katzir**[1] **Abhinav Nakarmi**[2] **Eyal Ronen**[1] **Mahmood Sharif**[1] **Or Patashnik**[1]

[1]Tel Aviv University   [2]University of Michigan

## Abstract

With the rapid adoption of diffusion models for visual content generation, proving authorship and protecting copyright have become critical. This challenge is particularly important when model owners keep their models private and may be unwilling or unable to handle authorship issues, making third-party verification essential. A natural solution is to embed watermarks for later verification. However, existing methods require access to model weights and rely on computationally heavy procedures, rendering them impractical and non-scalable. To address these challenges, we propose *NoisePrints*, a lightweight watermarking scheme that utilizes the random seed used to initialize the diffusion process as a proof of authorship without modifying the generation process. Our key observation is that the initial noise derived from a seed is highly correlated with the generated visual content. By incorporating a hash function into the noise sampling process, we further ensure that recovering a valid seed from the content is infeasible. We also show that sampling an alternative seed that passes verification is infeasible, and demonstrate the robustness of our method under various manipulations. Finally, we show how to use cryptographic zero-knowledge proofs to prove ownership without revealing the seed. By keeping the seed secret, we increase the difficulty of watermark removal. In our experiments, we validate NoisePrints on multiple state-of-the-art diffusion models for images and videos, demonstrating efficient verification using only the seed and output, without requiring access to model weights.

## 1 Introduction

Generative diffusion and flow models (Ho et al., 2020; Song et al., 2020; Lipman et al., 2022) have rapidly transformed visual content creation, enabling the synthesis of high-quality images and videos from simple text prompts (Rombach et al., 2021; Saharia et al., 2022; Ramesh et al., 2022). While these models open new creative opportunities, they also raise pressing questions of copyright, authorship, and provenance (Zhu et al., 2018; Yu et al., 2021; Liu et al., 2023). In particular, proving authorship of generated content is essential for creators who wish to protect their work, establish ownership, or resolve disputes over originality (Arabi et al., 2025; Huang et al., 2025). This challenge is particularly pressing for independent creators and smaller organizations, who lack the trusted infrastructure of major AI providers and therefore require alternative mechanisms, such as watermarking, to prove that content was generated by their models.

Watermarking has emerged as a promising direction for enabling authorship verification in generative models. In this setting, a watermark refers to a verifiable signal that links generated content to its origin. Most existing methods achieve this either by embedding artificial patterns into the output or by recovering hidden information through inversion of the generation process (Gunn et al., 2024; Arabi et al., 2025; Yang et al., 2024b; Wen et al., 2023; Ci et al., 2024). However, these approaches often require access to the model weights and inference code, which may not be available when the model is proprietary or privately fine-tuned. Others modify the generation process in ways that alter the output distribution, or rely on computationally expensive inversion procedures, making verification impractical at scale. These limitations hinder the adoption of watermarking in scenarios where efficient and model-agnostic solutions are most needed.

In this work, we propose *NoisePrints*, a lightweight watermarking scheme that does not embed additional signals or alter the generation process, thereby preserving the original output distribution.

Instead, we leverage the random seed that initializes the diffusion process as a proof of authorship. Our key observation is that the initial noise derived from a seed is highly correlated with the generated visual content (Łukasz Staniszewski et al., 2025). This property enables verification by directly checking the correlation between the initial noise and the generated content, without requiring access to the diffusion model or costly inversion procedures. To secure this construction, we incorporate a one-way hash function into the noise sampling process, which makes it infeasible to recover a valid seed from the content. Moreover, we employ cryptographic zero-knowledge proofs to establish ownership without exposing the seed, thereby increasing the difficulty of watermark removal. Finally, we design a protocol for resolving disputes over authorship claims that handles both watermark injection attempts and geometric transformations of the original content.

To assess the reliability of NoisePrints, we evaluate its security and robustness. We show that the probability of randomly sampling a seed that produces noise correlating with a given image above the verification threshold is vanishingly small, and provide intuition for why such correlations persist. We further examine robustness under a wide range of attacks, including post-processing operations, geometric transformations, SDEdit-style regeneration (Meng et al., 2021), and DDIM inversion (Song et al., 2022), and introduce a dispute protocol that complements the verification protocol. Finally, we compare our approach with existing watermarking methods, highlighting efficiency, robustness, and practicality, and discuss extensions such as zero-knowledge verification for real-world deployment.

Our results establish seed-based watermarking as a practical and robust solution for proving authorship in diffusion-generated content. The method requires no changes to the generation process, preserves output quality, and remains reliable across diverse models and adversarial conditions, providing creators with a lightweight tool to assert ownership in the growing landscape of generative media.

## 2 RELATED WORK

Watermarking in diffusion models can be organized along three design axes: timing (post-hoc vs. during sampling), location (pixels, latents/noise, or model parameters), and verification (direct decoding vs. inversion). We focus on sampling-time watermarking in the noise/latent space, embedding the mark directly in the generation trajectory. Unlike most prior works, which depend on inversion, our approach achieves lightweight, inversion-free verification without requiring access to model weights.

**Post-hoc Watermarking** Post-hoc methods embed a watermark into an image after it is generated. Early approaches used frequency-domain perturbations or linear transforms (Cox et al., 1997; O'Ruanaidh & Pun, 1997; Chang et al., 2005), while more recent works train deep networks to hide and extract invisible signals (Zhu et al., 2018; Zhang et al., 2019; Tancik et al., 2020). These methods are simple to deploy, since they require no changes to the generative model. However, they are fragile and can be defeated by regeneration or steganalysis attacks (Zhao et al., 2024; Yang et al., 2024a).

**In-generation Watermarking** Another line of work modifies the generative pipeline itself, often by fine-tuning the model so that watermarks are embedded directly into the produced images (Zhang et al., 2019; Zhao et al., 2023; Fernandez et al., 2023; Lukas & Kerschbaum, 2023; Cui et al., 2024; Sander et al., 2025; Zhang et al., 2024). These methods achieve strong detectability under common image transformations, but incur non-trivial training cost and require model weights, limiting portability and practical deployment.

Closer to our approach are methods that manipulate the noise used to initialize the denoising process, thereby embedding the watermark in the noise. Detection relies on inversion (e.g., DDIM inversion (Song et al., 2022)) to estimate the noise that generated the image and check whether it contains the watermark. Early schemes embedded patterns in the noise, but this introduced distributional shifts (Wen et al., 2023). Later works addressed this either by refining the embedded patterns (Ci et al., 2024; Yang et al., 2024b) or by sampling the noise with pseudorandom error-correcting code (Gunn et al., 2024; Christ et al., 2024). Another recent approach (Arabi et al., 2025) treats initial noises as watermark identities and matches inverted estimates against a database, using lightweight group identifiers to reduce search cost while still relying on inversion.

While these methods avoid the cost of training or fine-tuning a generative model, they transfer the computational overhead to the verification stage, since inversion requires repeatedly applying the diffusion model. This becomes especially prohibitive for high-dimensional data such as video. Dependence on inversion also limits their applicability to few-step diffusion models, where accurate recovery of the initial noise can be more challenging (Garibi et al., 2024; Samuel et al., 2025). Finally,

verification requires access to the generative model itself, which becomes restrictive if the model is private and its owner is either untrusted or unwilling to handle detection. Our method avoids these drawbacks: it neither embeds patterns nor alters the generation process, making it fully distortion-free, and it avoids reliance on inversion, enabling lightweight verification at scale.

**Attacks, Steganalysis, and Limits**  Regeneration attacks, which regenerate a watermarked image through a generative model to wash out the hidden signal while preserving perceptual quality, can reliably erase many pixel-space watermarks, challenging post-hoc approaches (Zhao et al., 2024). For content-agnostic schemes that reuse fixed patterns, including noise-space marks, simple steganalysis by averaging large sets of watermarked images can recover the hidden template, enabling removal and even forgery in a black-box setting (Yang et al., 2024a). At a more fundamental level, impossibility results show that strong watermarking, resistant to erasure by computationally bounded adversaries, is unattainable under natural assumptions, underscoring the need to specify precise threat models and robustness criteria (Zhang et al., 2025).

## 3 METHOD

### 3.1 PRELIMINARIES

We present our method in the context of latent diffusion models (LDMs) (Rombach et al., 2021; Podell et al., 2024; Labs, 2024), which have become the standard in recent diffusion literature. LDMs generate content by progressively denoising a latent and decoding it into pixel space with a variational autoencoder (VAE). An LDM consists of (i) a diffusion model that defines the denoising process, and (ii) a VAE $(E, D)$, where $E$ encodes images into latents and $D$ decodes latents back into pixels.

Generation begins from a seed $s$. To ensure that the noise generation process cannot be adversarially manipulated to yield a targeted noise initialization, we first apply a fixed cryptographic hash $h(s)$ and use the result to initialize the PRNG. We require $h$ to be deterministic, efficient, and cryptographically secure (collision resistant, pre-image resistant, and producing uniformly distributed outputs). The PRNG produces Gaussian noise $\varepsilon(h(s)) \sim \mathcal{N}(0, I)$, which the diffusion model iteratively denoises into a clean latent $z_0$. For the denoising process, we use deterministic samplers. Finally, the decoder $D$ maps $z_0$ to the output $x$, such that the seed $s$ uniquely determines the result via its hashed initialization of the PRNG.

In practice, the VAE is often public and reused across models (e.g., Wan (Wan et al., 2025) and Qwen-Image (Wu et al., 2025) share a VAE, and DALL·E 3 (Betker et al., 2023) uses the same VAE as Stable Diffusion (Rombach et al., 2021)). In this work we consider both diffusion and flow models. Both start from Gaussian noise $\varepsilon(h(s))$ and define a trajectory to a clean latent, making our verification framework applicable in either case, as demonstrated on Stable Diffusion (Rombach et al., 2021) and Flux (Labs, 2024). We assume the diffusion model is private and inaccessible to verifiers, while the VAE is accessible to them, allowing verifiers to embed candidate content into the shared latent space. Notably, we do not assume the VAE weights are publicly shared, and only assume black-box access to the VAE encoder. For brevity, we refer to the diffusion model simply as the model.

### 3.2 THREAT MODEL

We consider a setting where a generative model is controlled by a model owner who keeps the weights private and may expose the model only through a restricted interface (e.g., API access). The owner may be a small organization or even a private individual, and is not necessarily a fully trusted entity. Content can be generated either by the owner directly, or by a user who queries the model through the API. In both cases, the party who generated the content may later wish to prove authorship of the output without requiring access to the model itself. Since the model owner may not be willing, able, or trusted to handle authorship issues (now or in the future), the responsibility for verification is delegated to an independent third party. The verifier is the only trusted party for handling authorship claims, and its role is to execute the public verification procedure. The model weights remain private and are never shared.

To enable authorship verification, the content producer records the seed $s$ used to initialize the sampling procedure. The generated content $x$ is public, but $s$ remains secret until the producer wishes to prove authorship. At that point, the producer provides the pair $(x, s)$ to a verifier. The verifier can then check this claim using only public primitives (PRNG specification, encoder $E$, and

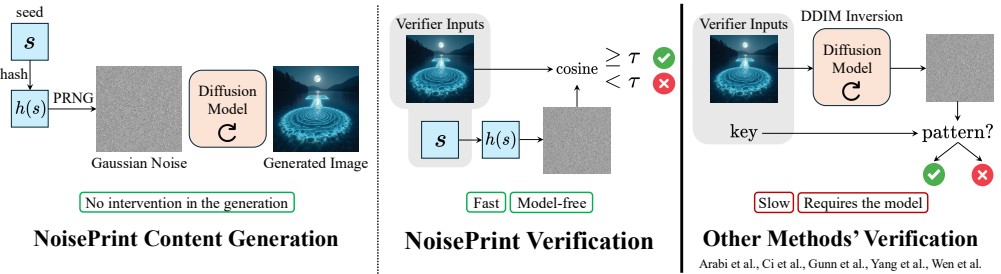

Figure 1: *NoisePrint* introduces no intervention in the generation process and therefore does not alter the distribution of generated images. For verification, we compare the noise derived from the seed with the given image. In contrast to other approaches that rely on DDIM inversion and compare the predicted initial noise to a key (i.e., a pre-embedded watermarking pattern) to decide authorship, our method is lightweight and model-free.

threshold calibration), without access to the model itself. This property ensures that verification is both lightweight and model-free, avoiding the need to share private weights.

It is important that the seed $s$ does not leak to the public during verification, as it could potentially allow an adversary to claim ownership over the content in the future or execute more targeted and effective adversarial attacks. In order to mitigate this risk, it would be beneficial if $s$ is not revealed even to the verifier. To support this, the scheme can be extended with zero-knowledge proofs that establish ownership without revealing $s$, which we recommend for practical deployments. For clarity, we first present our method without this extension.

We consider an adversary that knows the generated content $x$, and all public primitives: PRNG specification, encoder $E$, and verification threshold $\tau$. The adversary does not know the weights of the diffusion model and the seed $s$ used by the rightful owner. The adversary may pursue two goals:

- **Watermark Removal.** Modify $x$ into $\tilde{x}$ so that the correlation with the rightful owner's seed $s$ drops below the threshold $\tau$, making the content unverifiable.

- **Watermark Injection.** Produce an image $\tilde{x}$ that is visually similar to $x$, and a fake seed $s'$ such that $(\tilde{x}, s')$ passes verification, thereby claiming ownership of content similar to $x$.

An adversary may perform only removal (*removal-only*), only injection (*injection-only*), or a *combined* attack that both suppresses the original correlation with $s$ and establishes a correlation with a forged seed $s'$. To pursue the removal goal, we assume the adversary may employ the following types of attacks, all of which must preserve perceptual similarity to $x$: (i) basic image processing operations (e.g., compression, blur, resizing), (ii) diffusion-based image manipulation (e.g., SDEdit which reintroduces noise at intermediate denoising steps and DDIM inversion based optimization), or (iii) geometric transformations (e.g., rotations, crops).

These attack families follow prior work (Arabi et al., 2025; Gunn et al., 2024; Yang et al., 2024b) on robust watermarking and reflect both common manipulations that occur in practice and stronger generative edits that adversaries might attempt. Although we demonstrate robustness against one adversarial removal attack (DDIM inversion based optimization), we acknowledge that perfect robustness against adversarial, quality-preserving edits is unattainable (Zhang et al., 2025), and therefore scope our claims to practical robustness under bounded, perceptual-preserving manipulations. To measure the perceptual similarity between the original image and the attacked one, we use SSIM, PSNR, and LPIPS (Zhang et al., 2018).

### 3.3 NOISEPRINTS WATERMARKS

Our key observation, upon which we build our method, is that in diffusion and flow models the initial Gaussian noise $\varepsilon(s)$ leaves a persistent and surprisingly strong imprint on the generated content. Despite the high dimensionality of the space, the latent representation of the final image $x$ exhibits a significantly higher correlation with its originating noise $\varepsilon(s)$ than with an unrelated noise sample. A related observation was noted by Łukasz Staniszewski et al. (2025), though in a different context. We further discuss this phenomenon in Appendix A, where we relate it to optimal transport and propose an explanation for why such correlations naturally persist. This finding allows us to treat the initial noise as a natural watermark. By producing $\varepsilon(h(s))$ from the seed and measuring its correlation with $x$, we obtain a reliable authorship signal, which we call a *NoisePrint*. Unlike many prior watermarking approaches, NoisePrint does not alter the generative process and hence the output

distribution remains intact, *rendering the watermark completely distortion-free*. Figure 1 gives an overview of our method alongside a comparison to prior approaches based on inversion.

Next, we describe our verification protocol. This protocol remains robust under simple image processing and diffusion-based manipulations. To address more challenging cases such as geometric transformations and injection-only attacks, we further introduce a dispute protocol.

**Verification Protocol**    Let $E(x)$ denote the latent embedding of an image $x$ obtained using the public VAE encoder. For a given seed $s$, the initial Gaussian noise $\varepsilon(h(s))$ is produced deterministically by seeding the public PRNG (after hashing $s$) and drawing the required number of variates. We define the NoisePrint score as the cosine similarity between the embedded image and the noise:

$$\phi(x, s) \triangleq \frac{\langle E(x),\ \varepsilon(h(s)) \rangle}{\|E(x)\|_2\, \|\varepsilon(h(s))\|_2}. \tag{1}$$

A claim $(x, s)$ is verified by comparing $\phi(x, s)$ to a threshold $\tau$ calibrated to achieve a desired false positive rate under the null hypothesis that $E(x)$ and $\varepsilon(h(s))$ are independent. If $\phi(x, s) \geq \tau$, the verifier accepts the claim as valid. We summarize the verification protocol in Algorithm 1.

While this procedure is effective under simple image processing and diffusion-based manipulations, as demonstrated in Section 5.2, it does not address cases where the adversary applies geometric transformations or injects a watermark into an existing image. Geometric transformations can misalign the image embedding with its originating noise and therefore decrease the correlation, while injection attacks pose a challenge because an adversary may fabricate a different seed-image pair that also passes verification. To handle geometric transformations and injection-only attacks, we introduce a dispute protocol.

**Dispute Protocol**    We propose a dispute protocol for cases where an adversary claims ownership over a modified version of the original content, and passes the basic verification protocol due to injecting their watermark into this modified version. The protocol requires each claimant $i \in \{A, B\}$ to submit a triplet $(x_i, s_i, g_i)$ consisting of their content, their seed, and an optional transformation $g_i \in \mathcal{G}$ from a public family of transformations (e.g., rotations or crops). For a claim $(x, s)$ and a transformation $g \in \mathcal{G}$, we define the extended NoisePrint score:

$$\phi(x, s; g) \triangleq \frac{\langle E(g \cdot x),\ \varepsilon(h(s)) \rangle}{\|E(g \cdot x)\|_2\, \|\varepsilon(h(s))\|_2}. \tag{2}$$

The verifier then applies $g_i$ to the opponent's content and evaluates:

$$\phi(x_i, s_i; \mathrm{id}) \geq \tau \quad \text{and} \quad \phi(x_j, s_i; g_i) \geq \tau, \ \ j \neq i, \tag{3}$$

where $\mathrm{id}$ is the identity transformation. We refer to the first inequality as self check and the second as cross check. If one claimant satisfies both inequalities, that claimant is recognized as the rightful owner; if both or neither do, the dispute remains unresolved. The protocol is outlined in Algorithm 2.

The protocol resolves injection-only attempts. Suppose an adversary produces $\tilde{x}$ and a fake seed $s'$ such that $(\tilde{x}, s')$ passes the verification test, while the true NoisePrint from the rightful seed $s$ remains detectable. In the dispute, the rightful owner submits $(x, s, \mathrm{id})$ and passes both the self and cross checks. The injector, however, fails the cross check on $x$ with $s'$, since $s'$ is independent of $x$ under the null used to calibrate the threshold. Hence, injection without removal cannot overturn ownership, and any successful injector must also remove the true NoisePrint.

The dispute protocol also resolves geometric removal attempts. If an adversary applies some transformation $g$ to suppress the correlation of $(x, s)$, the rightful owner can recover alignment by submitting $(x, s, I(g))$, with $I(g) \in \mathcal{G}$ being the inverse transformation that would align the two. In doing so they would pass both checks, while the adversary cannot provide a valid seed for any geometric transformation $g \in \mathcal{G}$ of the original image. Note that it is not necessary for the inverse transformation $I(g)$ to fully recover the original image when applied on the transformed image, as long as the respective latents are re-aligned. See for example the set of transformations in Appendix H.

### 3.4    Zero-knowledge Proof

In this subsection, we provide a short background on zero-knowledge proof (ZKP) and describe the goal of our ZKP. Implementation details and benchmark results are provided in Appendix C.

Zero-knowledge proofs (ZKPs) allow a prover $P$ to convince a verifier $V$ that a statement is true without revealing to $V$ anything beyond its validity. Consider a public circuit $C$. Suppose a prover wants to convince a verifier that $y = C(s; x)$, where $y$ and $x$ are public and $s$ is a private witness known only to $P$. A ZKP lets $P$ produce a proof that convinces $V$ that $y$ was correctly computed as $C(s; x)$ for some $s$, without revealing $s$. In our case, all computation is performed over a finite field. Besides zero-knowledge, a ZKP must satisfy: (i) *Completeness:* if true, an honest prover can generate a proof accepted by the verifier (with high probability); and (ii) *Soundness:* if false, even a malicious prover cannot generate a proof accepted by the verifier (with high probability). For a more formal explanation of ZKPs, see (Thaler et al., 2022).

In our case, the private witness $s$ is the seed, and the public input $x$ is the image. The circuit $C$ uses $s$ to derive the initial noise, which is then used to compute an inner product with $x$. From this, the cosine similarity between the noise and the image is calculated. Finally, the circuit outputs 1 if the similarity exceeds the public threshold $\tau$, and 0 otherwise.

In addition, we use the ZKP to bind the proof to a specific user by partitioning the seed into two parts. The first part is a public string describing the image and ownership (e.g., "An image of a cat generated by the amazing cat company"), and the second part is a private secret random value. The concatenation of the public string and the private random value is used as input to the cryptographic one-way hash function $h$, whose output is then used to derive the initial noise for image generation. The resulting ZKP uses the string as a public input and is thus "bound" to the string and honest owner.

## 4 Security Analysis

The main security requirement in our setting is that it should be computationally infeasible for an adversary to forge a valid claim without access to the true seed. In the case of NoisePrints, this amounts to showing that it is extremely unlikely to find a random seed $s'$ such that the corresponding noise $\varepsilon(s')$ exhibits high correlation with a given image $x$. We emphasize that this analysis addresses only the probability of a random seed coincidentally passing verification, and does not cover manipulations of the content. Robustness against such attacks is evaluated empirically in Section 5.2.

**False positives under seed guessing**  Let $z = E(x) \in \mathbb{R}^d$ be the embedding of the candidate content, and let $\varepsilon \sim \mathcal{N}(0, I_d)$ be an independent random noise vector obtained from a random seed. The NoisePrint score is: $\phi = \langle z, \varepsilon \rangle / (\|z\|_2 \|\varepsilon\|_2)$. Without loss of generality we can assume that both $z$ and $\varepsilon$ lie on the unit sphere. Thus $\phi$ is simply the inner product between two independent random unit vectors in $\mathbb{R}^d$. In high dimensions, by the concentration of measure phenomenon, such vectors are almost orthogonal, hence their inner product is tightly concentrated around zero. The condition $\phi \geq \tau$ has a geometric interpretation: it means that the random noise $\varepsilon$ falls into a spherical cap of angular radius $\arccos(\tau)$ around $z$. In Appendix B we analyze this probability and show that:

$$\Pr[\phi \geq \tau] = \tfrac{1}{2} I_{1-\tau^2}\left(\tfrac{d-1}{2}, \tfrac{1}{2}\right) \leq \exp\left(-\tfrac{(d-1)}{2}\tau^2\right), \tag{4}$$

where $I_x(p, q)$ is the regularized incomplete beta function. The exponential decay in $d$ implies that the false positive probability becomes negligible in high-dimensional embeddings. This property naturally aligns with modern generative models: current image diffusion models already use thousands of dimensions, while video diffusion models employ embedding spaces an order of magnitude larger, making accidental collisions astronomically unlikely.

**Threshold selection**  Given a target false positive rate $\delta$, one can set the verification threshold $\tau$ as:

$$\tau = \sqrt{1 - a^*}, \quad \text{where } a^* \text{ solves } \tfrac{1}{2} I_a\left(\tfrac{d-1}{2}, \tfrac{1}{2}\right) = \delta. \tag{5}$$

In our case we target an extremely low rate of $\delta = 2^{-128}$, meaning an adversary would need to try roughly $2^{128}$ seeds to produce a false positive, which is computationally infeasible and provides cryptographic-level security. We find $a^*$ using a numerical solver.

## 5 Experiments and Results

This section presents an evaluation of our approach across various generative models. We begin by assessing the reliability of verification in the absence of attacks, measuring the true positive rate (TPR) at a fixed false positive rate (FPR). We then turn to robustness, examining how well NoisePrints withstand the range of attacks available to an adversary, and benchmarking our method against existing watermarking techniques. Since these baselines were designed under a different

threat model, we highlight two important distinctions: their verification requires access to the model weights, and it involves substantially higher computational cost. Additional experiments, analyses, and results are provided in Appendices E, F, H and L.

## 5.1 RELIABILITY ANALYSIS

We evaluate the reliability of verification with our approach across multiple models. Specifically, we generate images with Stable Diffusion 2.0 base (SD2.0, Rombach et al. (2021)), SDXL-base (Podell et al., 2024), Flux-dev, and Flux-schnell (Labs, 2024) using prompts from Gustavo (2022). For video generation, we use Wan2.1 (Wan et al., 2025) evaluated on a subset of prompts from VBench2.0 Zheng et al. (2025). For each generated image $x$, we compute its NoisePrint $\phi(x, s)$, where $s$ is the seed used to generate $x$, and report the mean and standard deviation. We then analytically determine a threshold per model for a fixed FPR of $2^{-128}$ (as in Section 4), and report the percentage of images that pass this threshold. Results are summarized in Table 1. NoisePrint values exceed the threshold by a large margin across all models, even at an extremely low FPR of $2^{-128}$. A single consistent outlier appears across three models, corresponding to a prompt discussed in Appendix G.

Table 1: Reliability analysis across different models. For each model, we report the latent image dimension $d$ (the dimension of the VAE latent space), the mean and standard deviation of the NoisePrint score $\phi(x, s)$, the analytically derived threshold $\tau$ for FPR $= 2^{-128}$, and the resulting pass rate (images detected as watermarked).

| Model | Latent Dim. ($d$) | Mean NoisePrint $\phi \pm$ Std | Threshold ($\tau$) | Pass Rate |
|---|---|---|---|---|
| SD2.0 | 16,384 | $0.482 \pm 0.088$ | 0.101739 | 1.00 |
| SDXL | 65,536 | $0.431 \pm 0.070$ | 0.051000 | 0.99 |
| Flux.1-schnell | 262,144 | $0.197 \pm 0.056$ | 0.025500 | 0.99 |
| Flux.1-dev | 262,144 | $0.202 \pm 0.055$ | 0.025500 | 0.99 |
| Wan2.1 | 1,297,920 | $0.0678 \pm 0.0247$ | 0.011460 | 1.0 |

## 5.2 ROBUSTNESS ANALYSIS

We analyze the robustness of our method using SD2.0 (Rombach et al., 2021), comparing it to prior works: WIND (Arabi et al., 2025), Gaussian Shading (GS) (Yang et al., 2024b), and Undetectable Watermark (PRC) (Gunn et al., 2024). We consider the attacks mentioned in Section 3.2. For each attack, we report the empirical true positive rate (TPR) as a function of the false positive rate (FPR). In addition, we measure TPR (at a fixed FPR) as a function of PSNR, LPIPS, and SSIM between the attacked image and the original. We also provide qualitative examples, visually demonstrating the effect of each attack on two sample images. Results are shown in Figures 2, 3, 10 and 11, with additional experiments on SDXL (Podell et al., 2024), Flux-schnell (Labs, 2024), and the video model Wan (Wan et al., 2025) in the Appendix (Figures 12 to 14 and 16 to 19).

Note that baseline methods require access to diffusion model weights, and their verification is substantially more computationally expensive as shown in Table 2. By replacing inversion with a lightweight cosine similarity, our method achieves an end-to-end verification speedup of $\times 14$–$\times 213$ over inversion-based baselines (WIND, PRC, GS), depending on the model.

**Basic Image Processing Attacks** We consider six common image corruptions, each applied at three severity levels: (i) brightness change (intensity multiplied by 2, 3, 4); (ii) contrast change (contrast multiplied by 2, 3, 4); (iii) Gaussian blur (Gaussian kernels of radius 2, 4, 6 pixels); (iv) Gaussian noise (additive noise with standard deviations 0.1, 0.2, 0.3); (v) compression (JPEG quality factors 25, 15, 10); and (vi) resize (down- and up-sampling with scale factors 0.30, 0.25, 0.20).

As shown in Figures 2, 10 and 11, our method matches or outperforms prior methods, achieving TPR above 0.9 at the lowest FPR ($2^{-128}$) for attacked images that retain reasonable perceptual similarity

Table 2: Runtime of different components for verifying various watermarking methods. All methods require one VAE encode. Baselines (WIND, PRC, GS) additionally perform inversion, while our method replaces it with a cosine similarity. Results are mean $\pm$ standard deviation over multiple runs on a single RTX 3090 GPU.

| Model | VAE Encode (all) | Inversion (WIND, PRC, GS) | Cosine Similarity (**Ours**) |
|---|---|---|---|
| SD2.0 (50 steps) | $0.037 \pm 0.004$ s | $3.234 \pm 0.075$ s | $0.182 \pm 0.045$ ms |
| SDXL (50 steps) | $0.152 \pm 0.007$ s | $12.704 \pm 0.303$ s | $0.090 \pm 0.018$ ms |
| Flux-dev (20 steps) | $0.158 \pm 0.007$ s | $33.594 \pm 0.245$ s | $0.098 \pm 0.005$ ms |
| Flux-schnell (4 steps) | $0.155 \pm 0.006$ s | $6.673 \pm 0.055$ s | $0.100 \pm 0.011$ ms |
| Wan2.1-1.3B (25 steps) | $6.463 \pm 0.102$ s | $91.473 \pm 0.164$ s | $0.097 \pm 0.010$ ms |

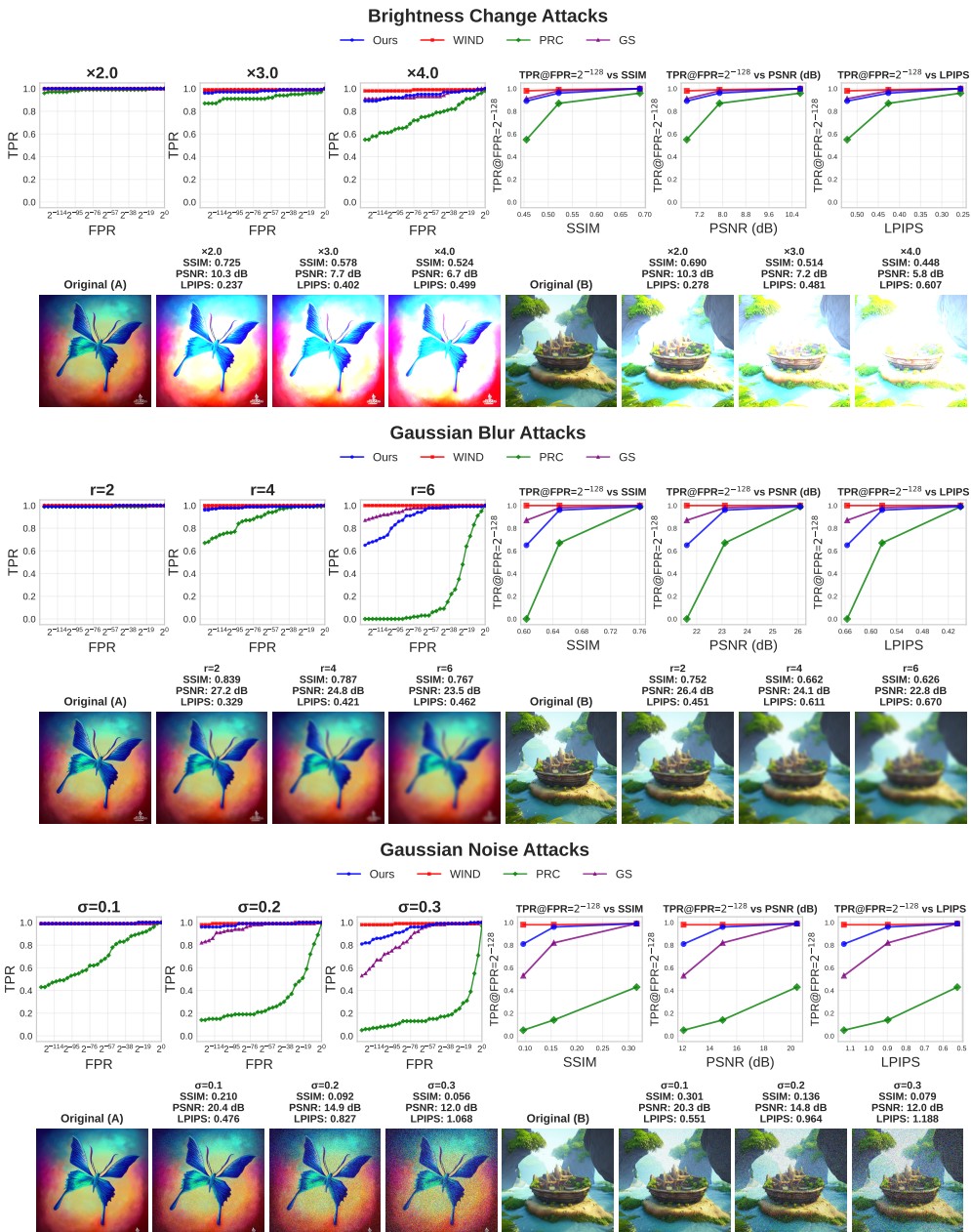

Figure 2: Robustness of different methods against common post-processing attacks. We evaluate brightness changes (top), Gaussian blur (middle), and Gaussian noise (bottom) at varying levels of severity.

and quality. Under severe degradations such as Gaussian blur ($r = 6$) and Gaussian noise ($\sigma = 0.3$), WIND appears more robust, with TPR near 1.0. However, at these corruption levels the images are heavily distorted and diverge from the outputs of a well-trained model (see sample images), making robustness in this regime less meaningful. Even at milder corruption levels our method maintains TPR above 0.9, though artifacts remain evident. For instance, with Gaussian noise at $\sigma = 0.2$, the images show strong artifacts and PSNR drops to 14.8-14.9, while TPR@FPR$= 2^{-128}$ is already close to 1.0, showing our method remains effective even when perceptual quality is compromised.

**Regeneration Attack** Following prior work (Zhao et al., 2024; Arabi et al., 2025), we evaluate diffusion-based regeneration (SDEdit-style) attacks (Meng et al., 2021; Nie et al., 2022) by adding Gaussian noise to the latent of a watermarked image and then denoising it back to a clean image. We test three noise levels: $0.2$, $0.4$, and $0.6$. To simulate the private-weights scenario, we apply SDEdit using a different base model, specifically SDXL, while the images were originally generated

with SD2.0. As shown in Figure 3, our method performs extremely well against this type of attack, surpassing prior methods. Importantly, regeneration attacks tend to preserve the perceptual quality of the original image, as evident from both the qualitative samples and the similarity metrics, making them a more realistic and concerning threat model than basic image corruptions.

**Inversion based adversarial attack**  While generic attacks such as image transformations or off-the-shelf regeneration can partially weaken the watermark signal, a stronger adversary could directly target our verification protocol, namely the correlation between noise and image. To explore this scenario, we introduce an optimization-based inversion attack that estimates the initial noise vector and deliberately decorrelates from it while preserving perceptual fidelity to the original image.

Specifically, the adversary estimates the initial noise $x_T$ used to generate the original image $x$ via DDIM inversion, and then optimizes the image latents $x_\theta$ (initialized to $x$) using the loss $L = \|x_\theta - x\|^2 + w \cdot \cos(x_\theta, x_T)$, where $w$ is a hyperparameter. We run 100 optimization steps with $w \in \{0.3, 0.4, 0.5\}$, where larger $w$ values encourage greater divergence from the original image. We consider two variations of the attack: (i) the attacker uses a different model (SD1.4) for initial noise estimation, and (ii) the attacker has access to the original generative model (SD2.0). DDIM inversion is performed with an empty prompt, 50 steps, and no classifier-free guidance (CFG).

As shown in Figures 3 and 10, both attack variations are significantly more effective than regeneration or image-transformation attacks. They preserve perceptual similarity to the original image, with only moderate degradation in quality. Nevertheless, our method outperforms all other baselines by a large margin, despite the attack being tailored to break our protocol, demonstrating resilience even under targeted adversarial conditions.

**Geometric Transformations**  We evaluate our method under geometric transformations, which disrupt the alignment between a generated image and its initial noise. Our dispute protocol addresses this by allowing each party to submit a transformation that re-aligns the opponent's image. Accordingly, we test performance when transformed images are restored using an estimated inverse transform. We focus on two transformation types, rotation and crop & scale, and find that, after re-alignment, 100% of images pass the verification threshold at FPR $= 2^{-128}$. See Appendix H for details.

## 6 Conclusion, Limitations, and Future Work

We presented NoisePrints, a method for authorship verification requiring only the seed and generated output, without access to diffusion model weights. Our approach does not alter the generation process and is hence distortion-free. Compared to prior watermarking methods, it is significantly more efficient, particularly for higher-dimensional models (e.g., video). We showed robustness under diverse manipulations, including diffusion-based attacks, where it outperforms existing methods.

Although our analysis focused on a specific threat model, our approach is broadly applicable. It is compatible with the owner-only setting of WIND (Arabi et al., 2025), supporting direct seed-image verification when the seed is known or serving as a lightweight pre-filter in their two-stage pipeline when it is not )see more details in Appendix I). More generally, our method can complement other watermarking schemes as a fast first-pass filter, reducing reliance on costly inversion or optimization in real-world deployments.

At the same time, our approach has limitations. It requires the verifier to be able to encode images through the model's VAE, which requires the cooperation of the model provider in case the VAE is private. It is unsuitable for real/fake detection, since adversarial patterns could be injected into real images to mimic correlation with a chosen noise. Our verification assumes a restricted set of geometric transformations, leaving open the possibility of stronger manipulations. Finally, like with other methods that perform verification against a key that relates to the original noise pattern, our method is not suitable for claiming authorship or tracing the origin of image or video variations that resemble the original only in their semantic content.

Looking forward, it would be interesting to extend our approach to real images, exploring how correlation-based methods could support real/fake detection in open-world scenarios. In this context, the spatial distribution of correlation may provide additional cues, for example by highlighting inconsistencies between foreground and background regions.

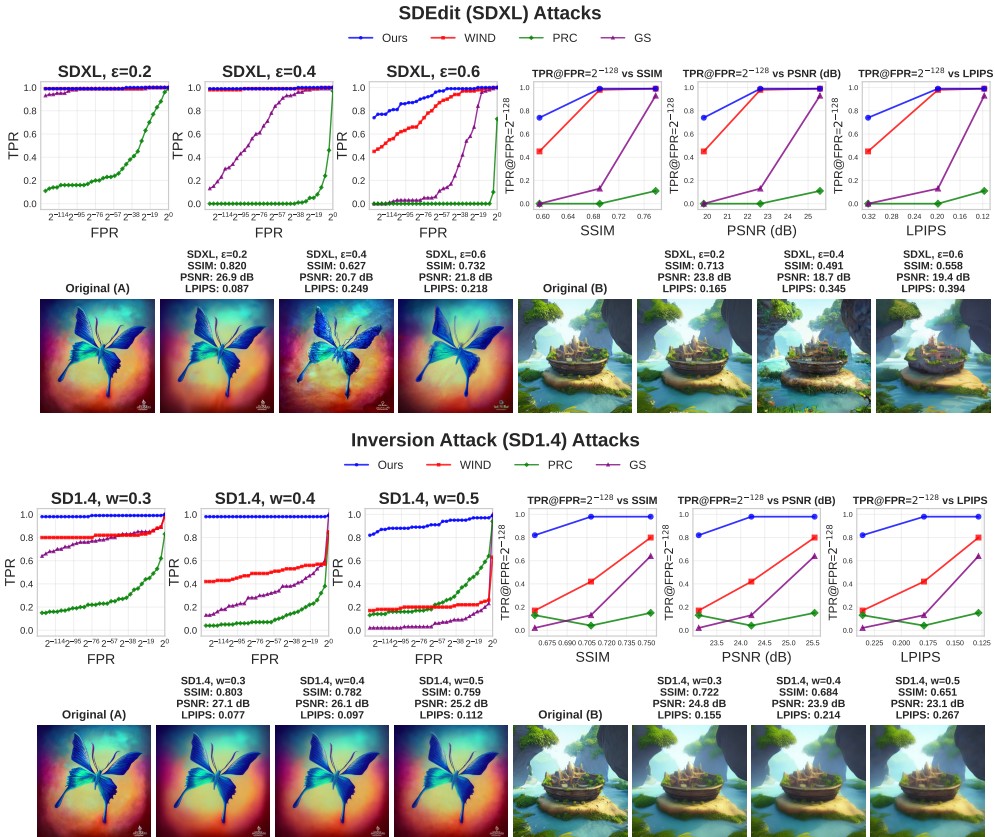

Figure 3: Robustness of different methods against regeneration and inversion attacks (using a different model).

## ETHICS STATEMENT

This work introduces a watermarking scheme for generative models aimed at improving authorship verification. Our method empowers creators, especially those without access to proprietary models, to establish ownership of their content. We believe this advances transparency and accountability in generative AI while minimizing risks of misuse. The approach does not alter the generation process, does not directly apply to real/fake detection, and is therefore unsuitable for monitoring or restricting legitimate content. We openly acknowledge that no watermarking system is perfectly robust and that our method should be viewed as a technical aid rather than a legal guarantee of authorship.

## REPRODUCIBILITY STATEMENT

We provide full details of our method, including the verification protocol, threat model, and zero-knowledge proof construction, in Section 3. All algorithms are described explicitly, and pseudocode for both verification and dispute protocols is included in Appendix D. Experimental settings, datasets, models, and evaluation metrics are specified in Section 5. Implementation details of the zero-knowledge proof are in Appendix C. Further implementation details are provided in Appendix K. To facilitate replication, we will publish code for reproducing all experiments and for generating the zero-knowledge proof. Together, these descriptions and resources should allow independent researchers to reproduce our results.

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

# APPENDIX

## A  OPTIMAL TRANSPORT DISCUSSION

**Optimal transport** studies the problem of moving probability mass from one distribution to another while minimizing a transport cost function. Given a source distribution $\mu$ and a target distribution $\nu$, the optimal transport map $T^*$ minimizes the expected cost $\mathbb{E}_{x \sim \mu}[c(x, T(x))]$ where $c(\cdot, \cdot)$ is the cost function, which is often set to be the quadratic cost $c(x, y) = \|x - y\|^2$. The optimal transport map provides the most efficient way to transform samples from the source to match the target distribution, which connects naturally to the generative modeling objective of transforming noise into data samples.

Khrulkov et al. (2022) demonstrate that the mapping between noise and data of the probability flow ODE of diffusion models coincides with the optimal transport map for many common distributions, including natural images. While not guaranteed in the general case (Lavenant & Santambrogio, 2022), they also provide theoretical evidence for the case of multivariate normal distributions.

Flow matching models are trained with conditional optimal transport velocity fields, and the learned velocity field is often simpler than that of diffusion models and produces straighter paths (Lipman et al., 2022). Liu et al. (2022) prove that rectified flow leads to lower transport costs compared to any initial data coupling for any convex transport cost function $c$, and recursive applications can only further reduce them.

By the identity $\|x - y\|^2 = \|x\|^2 + \|y\|^2 - 2\langle x, y \rangle$, decreases in transport cost correspond to increases in the dot product. The norm of high dimensional Gaussian noise samples concentrate tightly around $\sqrt{d}$, and assuming the target is a KL-regularized high dimensional VAE latent space, latent norms are encouraged to also have this property. Thus an increase in average dot product should translate to a near-proportional increase in average cosine similarity. We refrain from asserting a universal bound on the expected cosine for arbitrary targets, but on image/video data we empirically observe cosines that yield statistically decisive results with error probabilities compatible with cryptographic practice.

## B  EXACT SPHERICAL-CAP PROBABILITY FOR A GAUSSIAN VECTOR

Let $X \sim \mathcal{N}(\mathbf{0}, I_d)$ be a $d$-dimensional standard Gaussian and let $v \in \mathbb{R}^d$ be a unit vector. We are interested in the tail probability

$$\Pr\big[\cos(X, v) \geq a\big], \qquad a \in [-1, 1].$$

Because the Gaussian is rotationally invariant, we may assume $v = e_1$ without loss of generality.

**Theorem 1** (Exact spherical-cap probability). *For any $d \geq 2$ and $a \in [-1, 1]$,*

$$\boxed{\Pr\big[\cos(X, v) \geq a\big] \;=\; \tfrac{1}{2} I_{1-a^2}\Big(\tfrac{d-1}{2}, \tfrac{1}{2}\Big)} \tag{6}$$

*where $I_x(p, q)$ is the regularized incomplete beta function[1].*

*Proof.* Define the random direction $U := X/\|X\| \in S^{d-1}$, which is uniform on the sphere. Then

$$\cos(X, v) \;=\; \tfrac{Xv}{\|X\|} \;=\; U_1.$$

The first coordinate $U_1$ of a uniform point on $S^{d-1}$ has the density (Muller, 1959, Eq. (3.2))

$$f_d(t) \;=\; \frac{\Gamma\big(\tfrac{d}{2}\big)}{\sqrt{\pi}\,\Gamma\big(\tfrac{d-1}{2}\big)} \big(1 - t^2\big)^{\frac{d-3}{2}}, \qquad -1 < t < 1,$$

i.e. the $\text{Beta}\big(\tfrac{d-1}{2}, \tfrac{1}{2}\big)$ distribution mapped affinely from $[0, 1]$ to $[-1, 1]$. Integrating $f_d(t)$ from $a$ to 1 and expressing the result with the regularised incomplete beta function yields Equation 6.  □

---

[1] In SCIPY this is `scipy.special.betainc(p, q, x)`.

**Theorem 2** (Exponential bound). *For any $d \geq 2$ and $\tau \in [0, 1]$,*

$$\Pr\big[\cos(X, v) \geq \tau\big] \; \leq \; \exp\Big(-\tfrac{(d-1)}{2}\,\tau^2\Big).$$

*Proof.* Let $U := X/\|X\| \in S^{d-1}$, which is uniform on the sphere, and set $f(u) := \langle u, v \rangle$. The map $f : S^{d-1} \to \mathbb{R}$ is 1-Lipschitz (with respect to the geodesic or Euclidean metric restricted to the sphere) and has median 0 by symmetry. By Lévy's isoperimetric (concentration) inequality on the sphere (Ledoux, 2001, Ch. 2), for every $t \geq 0$,

$$\Pr\big[f(U) \geq t\big] \; \leq \; \exp\Big(-\tfrac{(d-1)}{2}\,t^2\Big).$$

Taking $t = \tau$ yields the claim. $\qquad\square$

## C  ZERO-KNOWLEDGE PROOF

In this section, we provide the implementation details and benchmark results of our zero-knowledge proof (ZKP).

### C.1  IMPLEMENTATION DETAILS

In our implementation, we had to overcome two main challenges:

1. ZKP proof systems currently do not allow for efficient proofs on floating-point number computations.

2. Proofs with input sizes required for our use case (vectors of sizes larger than $2^{18}$) are infeasible in our proof system due to high memory requirements both for the initial compilation of the circuit and for the proof generation.

We overcome these challenges by using fixed-point integers instead of floating points, and splitting the proof for the full vector derivation and inner product computation into smaller proofs of intermediate inner product computation (for vectors of size $\approx 700$) and then using another circuit to combine all of the intermediate values to find the cosine angle and check it against the threshold. This approach allows us to easily scale up our proofs to larger noise sizes as required for video generation.

We use the CirC (Ozdemir et al., 2022) toolchain to write our circuit in a front-end language called Z# and then compile it to an intermediate representation called R1CS. We then use CirC to produce a ZKP on the R1CS instance using the Mirage (Kosba et al., 2020) proof system. In particular, we use Woo et al. (2025)'s modified version of CirC.

As mentioned above, our circuit takes as private witness a seed $s$ and derives a vector $v_1$ of length $L$ (which in our implementation was chosen to be of size $266000 \approx 2^{18}$). The circuit then takes as public input a flattened image latent represented by a vector $v_2$ of size $L$. It then computes their dot product and their individual magnitudes. Finally, using these values, it computes the cosine angle $CA$ and checks if it is above a public threshold value $\tau$.

In more detail, the circuit uses private seed $s$ and a public seed $s_{\text{pub}}$ to derive the vector $v_1$ as follows. First, the circuit computes $p \leftarrow h(s \parallel s_{\text{pub}})$, where $h$ is a collision-resistant hash function. Then the circuit expands $p$ by iteratively applying a pseudorandom number generator (PRNG) to produce a stream of pseudorandom numbers.

These pseudorandom values are then used as inputs to a lookup table (Acklam, 2003) that approximates the inverse cumulative distribution function of the Gaussian distribution, thereby transforming the uniform pseudorandom numbers into Gaussian-distributed samples. The resulting values from the lookup table evaluation constitute the entries of $v_1$.

Unfortunately, our framework's memory requirements make computing a vector of size $\approx 2^{18}$ infeasible even in a server-class machine. To solve this issue, we construct two circuits instead of one. Our key idea is to make the first circuit prove the correctness of the dot product and magnitude using only $L/n$ entries at a time, for some $n$ such that $L/n$ is small enough. The prover can then generate $n$ proofs using this circuit to cover all $L$ entries. The second circuit then combines $n$ dot products

and $n$ magnitudes to produce a cosine angle to check if it is above a public threshold value $t$. As part of the proof, the prover commits to all intermediate inner product values. Both circuits verify these commitments to ensure that the intermediate values calculated and verified by the first circuit are also the ones used by the second circuit. Next, we describe both circuits in detail.

## C.2 ZKP CIRCUIT FOR COMPUTING DOT PRODUCT AND SQUARED MAGNITUDE

We provide the pseudocode of our first circuit in Figure 4. This circuit takes as input a private seed $s$ to derive a noise vector $(e_k)_{k \in [L/n]}$. To do so, along with $s$, it uses a public seed $s_{\text{pub}}$ (representing ownership information) and a public counter $c$ (identifying one of $n$ circuits) to compute $p \leftarrow h(s \parallel s_{\text{pub}} \parallel c)$. The circuit then iteratively computes $\text{PRNG}(p)$ to generate $g$ pseudorandom numbers. As numbers in CirC are elements in a prime field of size $\approx 255$ bits and we only need 33 random bits for our Gaussian noise sampling algorithm, each such pseudorandom number is divided into $k = 7$ parts, so that $(g \cdot k) = L/n$. They are then used to sample elements from the normal distribution using a lookup table ND which produces the noise vector $(e_k)_{k \in [L/n]}$. After the values are derived, the circuit calculates their inner product with the public input vector that represents the $L/n$-th portion of an image in the form of a vector $(v_k)_{k \in [L/n]}$. The circuit calculates both the dot product and the squared magnitude of $(e_k)_{k \in [L/n]}$. Finally, the circuit verifies that the public commitment $com$, combined with private randomness $r$, correctly commits to the dot product and squared magnitude (which values will be used by the second circuit). The commitment is instantiated using a hash function on the concatenation of the values. Since ZKP circuits operate over finite fields, negative integers cannot be represented directly, so the actual implementation uses an additional sign vector to encode them.

$$
\begin{aligned}
&\text{DPM}(c, s_{\text{pub}}, (v_k)_{k \in [L/n]}, com; s, r) : \\
&\quad dot\_prod = 0 \\
&\quad sq\_mag = 0 \\
&\quad p \leftarrow h(s \parallel s_{\text{pub}} \parallel c) \\
&\quad \textbf{for } i \in \{1, \dots, g\}: \\
&\quad\quad p \leftarrow \text{PRNG}(p) \\
&\quad\quad \text{//parse } p \text{ as } (p_\ell)_{\ell \in [k]}. \\
&\quad\quad \textbf{for } j \in \{1, \dots, k\}: \\
&\quad\quad\quad e_{(i,j)} \leftarrow \text{ND}(p_j) \\
&\quad\quad\quad dot\_prod \leftarrow dot\_prod + e_{(i,j)} \cdot v_{(i,j)} \\
&\quad\quad\quad sq\_mag \leftarrow sq\_mag + e_{(i,j)}^2 \\
&\quad\quad \textbf{endfor} \\
&\quad \textbf{endfor} \\
&\quad \textbf{assert}(com = \text{commit}(dot\_prod \parallel sq\_mag \parallel r)) \\
&\quad \text{return } 1
\end{aligned}
$$

Figure 4: Circuit for computing dot product and square of the magnitude. The circuit is instantiated with a function ND that on a random input simulates sampling an element from normal distribution.

## C.3 ZKP CIRCUIT FOR COMBINING ALL DOT PRODUCTS AND SQUARED MAGNITUDES

The pseudocode for the second circuit is shown in Figure 5. To start, the circuit takes as public input commitments $(com_i)_{i \in [n]}$ and as private inputs randomness $(r_i)_{i \in [n]}$, dot products $(dot\_prod_i)_{i \in [n]}$ and squared magnitudes $(sq\_mag_i)_{i \in [n]}$. It checks if all $com_i$ are valid. If so, using these values, the circuit calculates the final dot product $FDP$ and the final squared magnitude $FSM$, which represent all $L$ elements. Next, instead of computing the magnitude $mag$ of the entire noise vector from $FSM$, which requires a complex square root computation, the circuit takes it as a private input and checks if it is valid (which requires just a simple multiplication). Similarly, instead of computing the cosine angle $CA$, the circuit takes it as a private input and checks its correctness with the help of the public magnitude of the image vector $img\_mag$. Note that since a field does not recognize real numbers, we round down these values to the nearest integer and scale both cosine angle $CA$ and threshold $t$ to be 32-bit fixed-precision integers. Similarly to the earlier circuit, we handle negative values with an additional vector that represents the sign.

```
Combine((com_i)_{i∈[n]}, img_mag, t; (r_i)_{i∈[n]},
        (dot_prod_i)_{i∈[n]}, (sq_mag_i)_{i∈[n]}, mag, CA):
    FDP = 0
    FSM = 0
    for i ∈ {1, ..., n}:
        if com_i ≠ commit((dot_prod_i ‖ sq_mag_i); r_i):
            return ⊥
        FDP = FDP + dot_prod_i
        FSM = FSM + sq_mag_i
    endfor
    // verify magnitude of noise vector mag
    assert((mag)^2 <= FSM <= (mag + 1)^2)
    // verify cosine angle CA
    floor ← mag · img_mag · CA
    ceil ← mag · img_mag · (CA + 1)
    assert(floor <= FDP · 2^32 <= ceil)
    assert(CA > t)
    return 1
```

Figure 5: Circuit for combining all dot products and squared magnitudes.

### C.4 BENCHMARK RESULTS

We benchmarked our ZKPs to show that they are indeed efficient and practical. Our testbed is a machine equipped with an AMD Ryzen Threadripper 5995WX 1.8GHz CPU and 256GB RAM. The proof generation time for the first circuit is 765 ms (which can be run in parallel for all $n$ parts of the vector), whereas for the second circuit it is 920 ms. The proof verification times for the first and second circuits are 415 ms and 115 ms, respectively.

Since we use Mirage as our backend proof system, it produces a prover and verifier key required for proving and verifying, respectively. The prover key for both circuits is less than 200 MB, and the verifier key is less than 1 MB in both cases. The proof size is at most 356 bytes.

## D NOISEPRINT ALGORITHMS

---

**Algorithm 1:** Verification for NoisePrint

---

**Input:** content $x$, seed $s$, threshold $\tau$
**Public Primitives:** encoder $E$, PRNG spec, hash function $h$

**if** $\phi(x, s) \geq \tau$ **then return** Accept
**else return** Reject

---

---

**Algorithm 2:** Dispute Protocol

---

**Input:** claims $(x_A, s_A, g_A)$ and $(x_B, s_B, g_B)$
**Public Primitives:** encoder $E$, threshold $\tau$, PRNG spec, set of transforms $\mathcal{G}$, hash function $h$

**for** $i \in \{A, B\}$ **do**
    **if** $g_i$ *not provided* **then** $g_i \leftarrow$ id
    $\text{SELFPASS}(i) \leftarrow [\phi(x_i, s_i; \text{id}) \geq \tau]$
    $\text{CROSSPASS}(i) \leftarrow [\phi(x_j, s_i; g_i) \geq \tau], \; j \neq i$
    $\text{VALID}(i) \leftarrow \text{SELFPASS}(i) \wedge \text{CROSSPASS}(i)$

**if** $\text{VALID}(A)$ *and not* $\text{VALID}(B)$ **then return** A
**else if** $\text{VALID}(B)$ *and not* $\text{VALID}(A)$ **then return** B
**else return** Unresolved

---

## E   VAE EFFECT ON COSINE SIMILARITY

A practical consideration in our framework is that correlation is measured in latent space, whereas the generated content is ultimately observed in RGB space. This raises the question of whether decoding a latent to an image and then re-encoding it back into latent space affects the measured correlation. To evaluate this, we report the correlation values before and after a VAE decode-encode cycle, using the native VAE of each model. As shown in Table 3, the differences are minor across all tested models, indicating that the VAE introduces only negligible distortion and does not significantly affect the correlation.

| Model | Pre-VAE Mean $\pm$ Std | Post-VAE Mean $\pm$ Std |
|---|---|---|
| SD2.0 | $0.4922 \pm 0.0904$ | $0.4818 \pm 0.0876$ |
| SDXL | $0.4545 \pm 0.0598$ | $0.4283 \pm 0.0608$ |
| Flux.1-schnell | $0.2102 \pm 0.0535$ | $0.1989 \pm 0.0543$ |

Table 3: Cosine similarity of generated latents with original noise before and after passing through the VAE and back.

## F   CORRELATION QUALITATIVE ANALYSIS

Our method builds on the observation that the noise used to generate an image is highly correlated with the image itself. Figure 6 shows two examples, one from Flux (Labs, 2024) and one from SDXL (Podell et al., 2024), with spatial correlation maps smoothed by a Gaussian filter. Regions exceeding a predefined threshold are highlighted by an overlaid mask. As can be seen, the correlation is stronger in the foreground regions. We hypothesize that this effect arises from sharper structures and richer textures in foreground regions, where high-frequency details are more directly influenced by the noise, whereas smoother backgrounds dilute the signal.

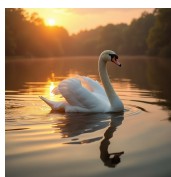 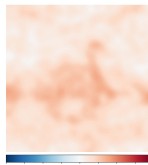 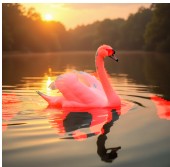 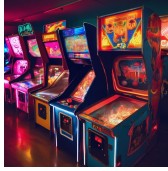 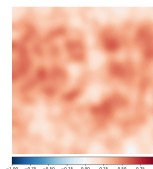 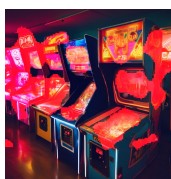

Figure 6: Spatial correlation between initial noise and the generated image latents. Left: Flux-dev, right: SDXL.

## G   FAILURE EXAMPLE

We observed a failure case with a specific prompt ("concept art of a minimalistic modern logo for a European logistics corporation"). For 2 out of the 3 models tested, the generated images had exceptionally low entropy and contained large uniform regions, making it much more difficult to retain a detectable watermark. In both SDXL and Flux.1-schnell, the resulting correlation fell below the threshold chosen for a $2^{-128}$ false positive rate, despite being generated by the claimed seed (Figure 7). A related result by Łukasz Staniszewski et al. (2025) demonstrates that DDIM inversion tends to produce latents that

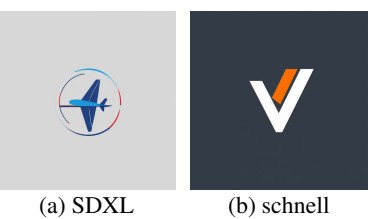

(a) SDXL          (b) schnell

Figure 7: Failure cases.

more significantly deviate from the original noise vector that was used to generate the image in parts of the latents that correspond to plain areas in the image. While such cases are rare, they highlight that verification may fail in low-variance generations. Importantly, this can be anticipated, and users can be warned at generation time if the output falls into this regime.

## H  GEOMETRIC TRANSFORMATIONS ATTACK

We next provide more details about the experiment that showed robustness to geometric attacks (Section 5.2, last paragraph).

As mentioned earlier, we consider two transformation types: rotation and crop & scale. For rotation, each image is rotated by a random angle in the range $[-45, 45]$ degrees. For crop & scale, the image is cropped at a random location with a crop factor in $[0.6, 0.9]$, and then rescaled to its original size. In both cases, the applied transformation is estimated using OpenCV's `estimateAffinePartial2D` function, and its inverse is used to re-align the image. To account for potential misalignment at the borders, we compute a transform-derived mask that restricts the cosine similarity calculation to the overlapping spatial region (see Figure 8).

Given a set of images, we apply these attacks and report the mean and standard deviation of the NoisePrint score, as well as the percentage of images that pass the verification threshold at FPR $= 2^{-128}$. As shown in Table 4, both rotation and crop & scale transformations are accurately estimated in all cases, resulting in 100% of images passing the verification threshold.

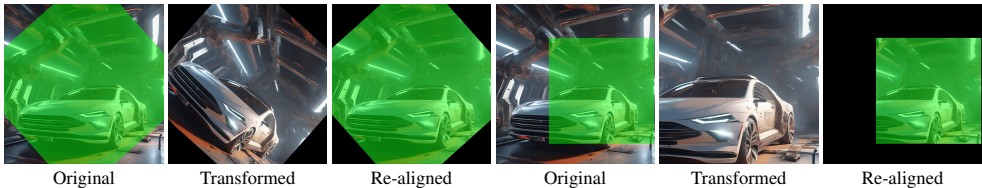

| Original | Transformed | Re-aligned | Original | Transformed | Re-aligned |

Figure 8: Estimation and alignment of geometric attacks. In green: the masked area used for cosine similarity.

Table 4: Quantitative results under geometric transformations. We report the mean and standard deviation of the NoisePrint score $\phi$ and the pass rate at FPR $= 2^{-128}$ for both rotation and crop & scale transformations. In all cases, the transformations are accurately estimated and every image passes the threshold.

| Transform | Mean NoisePrint $\phi\pm$ Std | Pass Rate |
|---|---|---|
| Rotation | $0.3825 \pm 0.0648$ | 1.0 |
| Crop & Rescale | $0.4191 \pm 0.0649$ | 1.0 |

## I  TRACEABILITY RESULTS

We evaluate NoisePrints in the traceability setting presented in WIND Arabi et al. (2025), where the model owner embeds a unique watermark per generated image to subsequently verify whether a given image was generated by their model and identify the specific key that was embedded during generation from a pool of keys. Table 5 reports detection rates for our method and WIND across 100 images matched against 100,000 different seeds under various corruptions and adversarial attacks using Stable Diffusion 2.0. Both methods demonstrate high robustness under most corruptions; however, our method exhibits superior resilience to the DDIM inversion-based adversarial attack.

Table 5: Tracability results: Fraction of images matching with the correct noise sample out of 100,000 (Stable Diffusion 2.0) **Attack legend:** Bright - Brightness $\times 3$, Contrast - Contrast $\times 3$, Blur - Gaussian Blur $r = 4$, Noise - Gaussian Noise $\sigma = 2$, Inv - DDIM inversion using Stable Diffusion 1.4 with $w = 0.4$, JPEG - JPEG compression $Q = 25$, Resize - Resize $\times 0.25$, SDEdit - SDEdit with SDXL, $\epsilon = 0.6$.

| Metric | Clean | Bright | Contrast | Blur | Noise | Inv | JPEG | Resize | SDEdit |
|---|---|---|---|---|---|---|---|---|---|
| Ours | 1.000 | 0.990 | 1.000 | 0.990 | 1.000 | 0.980 | 0.990 | 1.000 | 1.000 |
| WIND | 1.000 | 0.990 | 1.000 | 1.000 | 1.000 | 0.560 | 1.000 | 1.000 | 0.980 |

## J    COSINE SIMILARITY DISTRIBUTION

In Figure 9 we visualize the cosine similarity distribution of generated images with their respective Initial noises and of generated images with random Gaussian noises. This illustrates the large gap that allows for a clear separation between the two distributions.

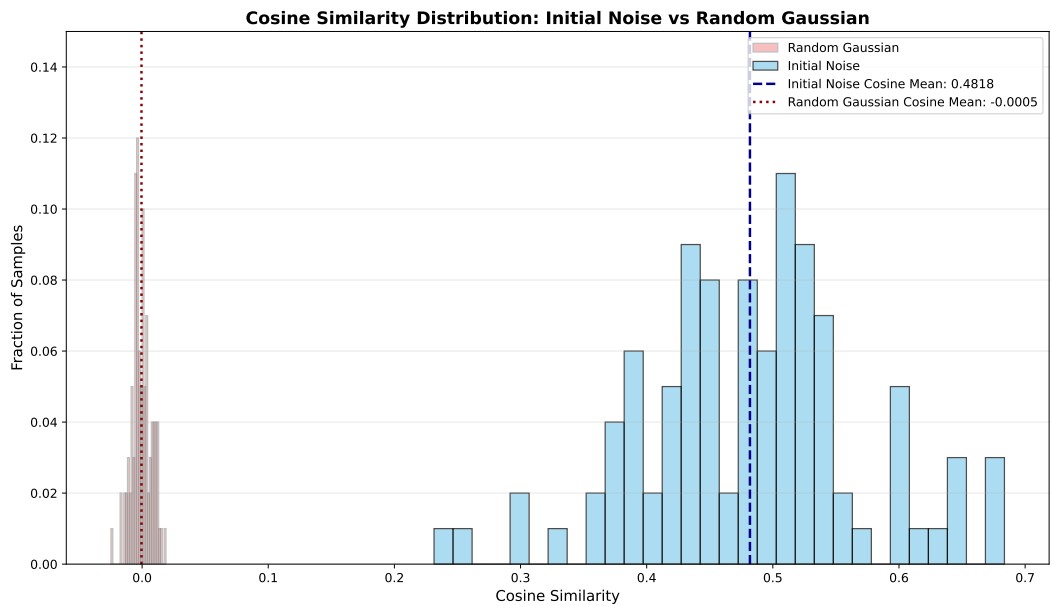

Figure 9: Histograms of cosine similarity between initial noise and generated images and cosine similarity between random noise and generated images (Stable Diffusion 2.0).

## K    IMPLEMENTATION DETAILS

For most of our experiments, we use `torch.randn` as the PRNG, passing a `torch.Generator` that is initialized with the seed. For the zero-knowledge proof implementation, we use a simple Linear Congruential Generator to get pseudorandom numbers, which are then transformed into Gaussian-distributed samples as explained in Appendix C. We use an implementation of Poseidon (Grassi et al., 2021) as our one-way hash function.

## L  ADDITIONAL ROBUSTNESS RESULTS

We provide additional robustness results for our method across different models:

1. Figure 10 reports results on SD2.0 under our inversion attack, where the model used for performing inversion is the same as the one used for image generation (SD2.0).

2. Figure 11 presents additional results on SD2.0 with basic corruption attacks.

3. Figures 12 and 13 shows results on SDXL with basic corruption attacks.

4. Figure 14 provides results on SDXL under SDEdit and inversion attacks, with SDXL also used to perform the attacks.

5. Figure 15 provides results on SDXL under the imprint removal attack presented in Müller et al. (2025). In this attack, the initial noise $x_T$ that was used to generate the original image latents $x$ is estimated with DDIM inversion using a proxy model (Stable Diffusion 2.0). Then, the image latents $x_\theta$ (initialized to $x$) are optimized such that the resulting latents from DDIM inversion stray away from this initial noise estimation of $x_T$. The optimization happens through gradient descent, minimizing the loss $L = \|I_{0 \to T}(x_\theta) - (-x_T)\|^2$, with $I_{0 \to T}(x_\theta)$ being the inversion result when starting from $x_\theta$. We evaluate the method's effectiveness after 30, 40, and 50 optimization steps. As can be seen, this attack is highly effective against watermarking methods that rely on inversion for detection, and is able to cause detection rates to significantly drop with only minor degradations to image fidelity. In contrast, our method displays very high robustness against this attack, since it does not rely on inversion and is less affected by the method's adversarial optimization goal.

6. Figures 16 and 17 presents results on Flux-schnell with basic corruption attacks. Note that Flux-schnell is a few-step model operating with only four denoising steps. Accurate inversion is more challenging in such models, making our inversion-free approach a significant advantage.

7. Figure 18 shows results on Flux-schnell under SDEdit and inversion attacks, with SDXL used to perform the attacks.

8. Figure 19 provides results on the video model Wan, where we adapt image attacks to the video domain. Our method demonstrates strong robustness on video while remaining efficient. As shown in Table 2, relying on correlation rather than inversion is particularly beneficial for video due to its high dimensionality.

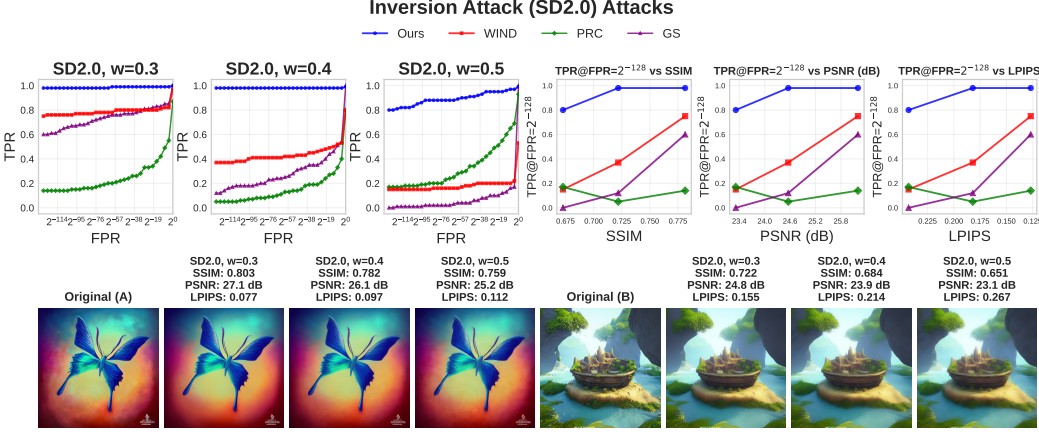

Figure 10: SD2.0: Comparing robustness of different watermarking methods against inversion attack.

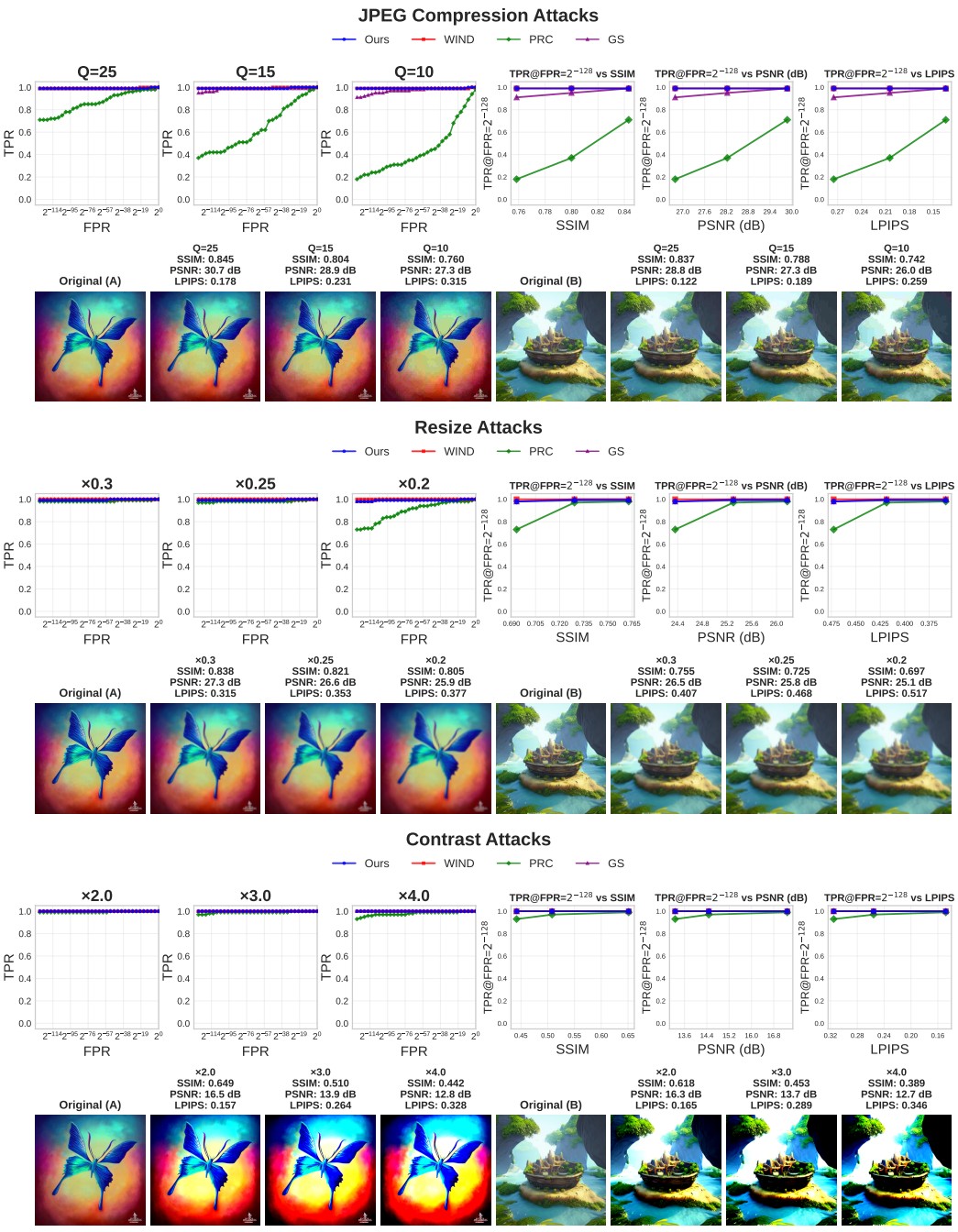

Figure 11: SD2.0: Comparing robustness of different watermarking methods against additional basic corruption attacks.

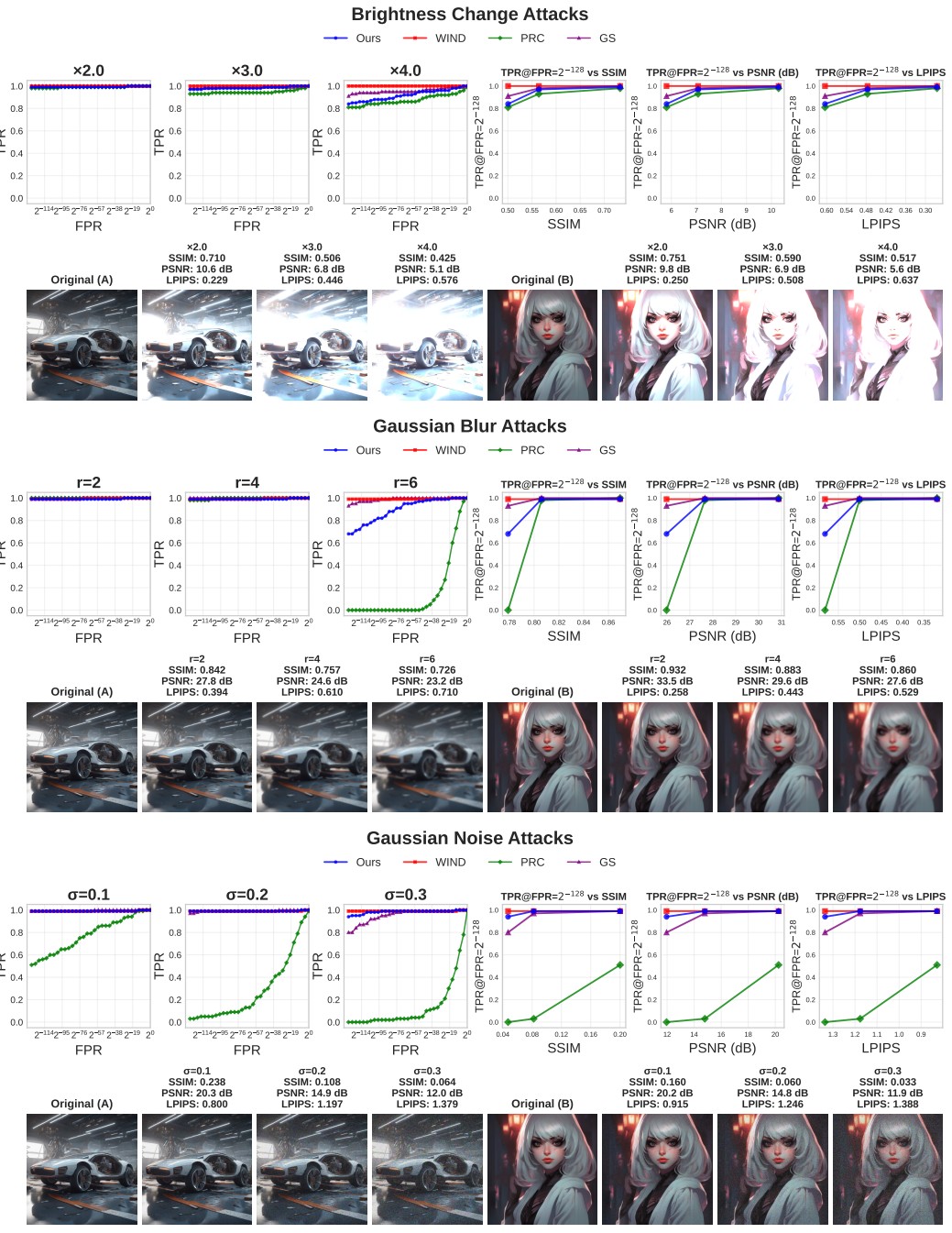

Figure 12: SDXL: Evaluating robustness against basic corruption attacks.

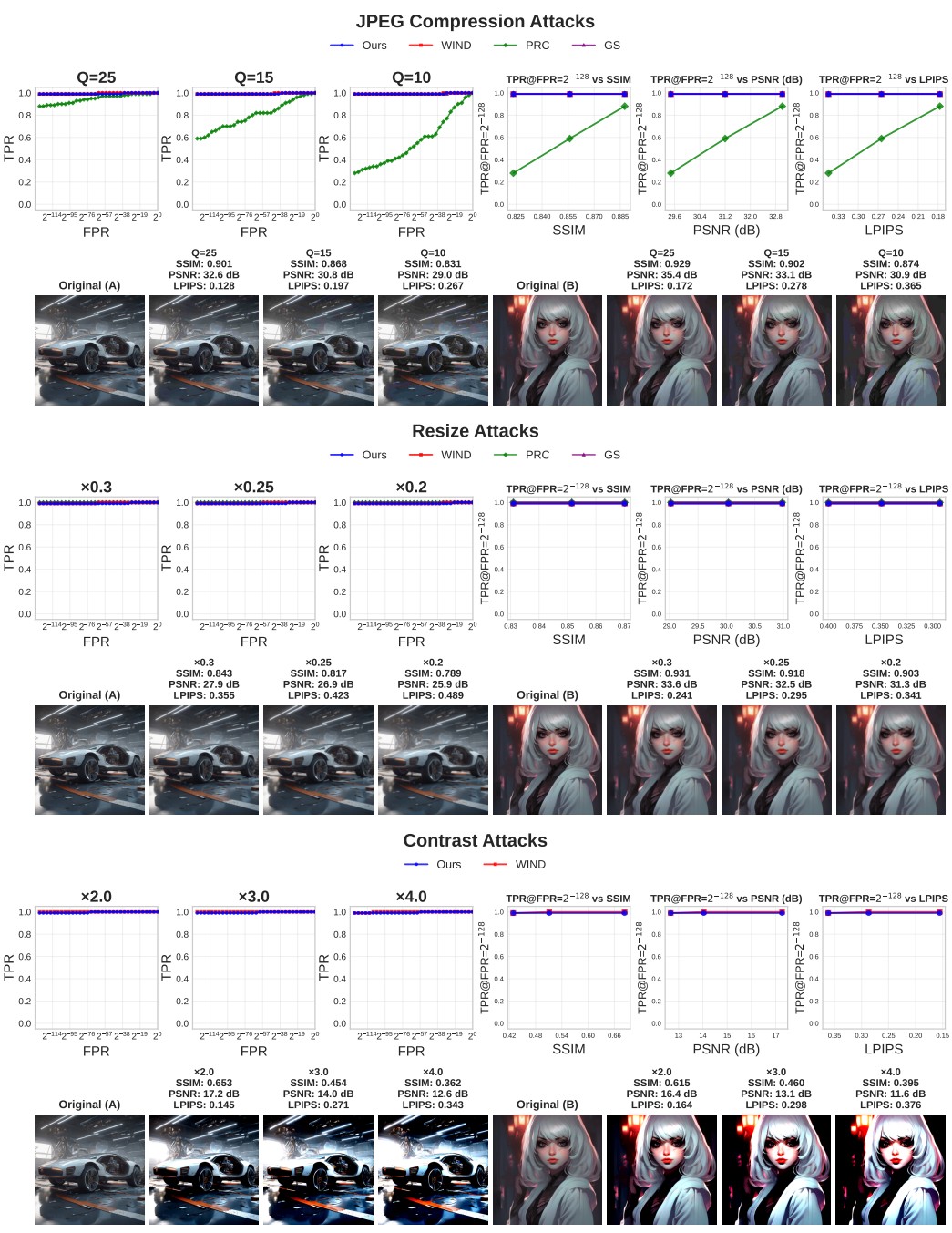

Figure 13: SDXL: Evaluating robustness against additional basic corruption attacks.

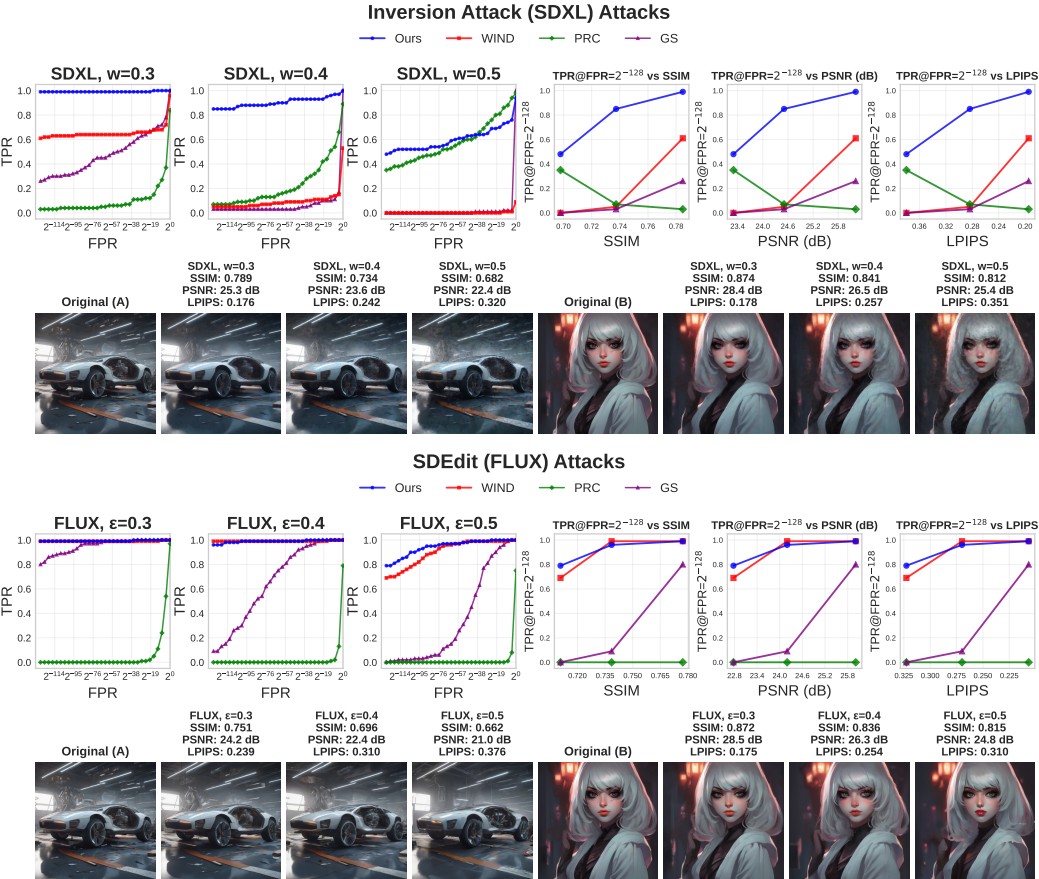

Figure 14: SDXL: Evaluating robustness against SDEdit and inversion attacks.

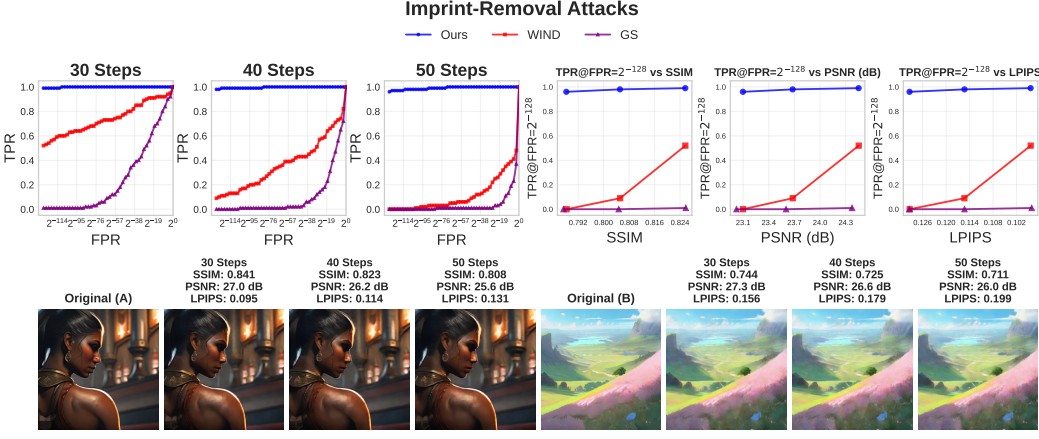

Figure 15: SDXL: Evaluating robustness against Imprint Removal attack of Müller et al. (2025).

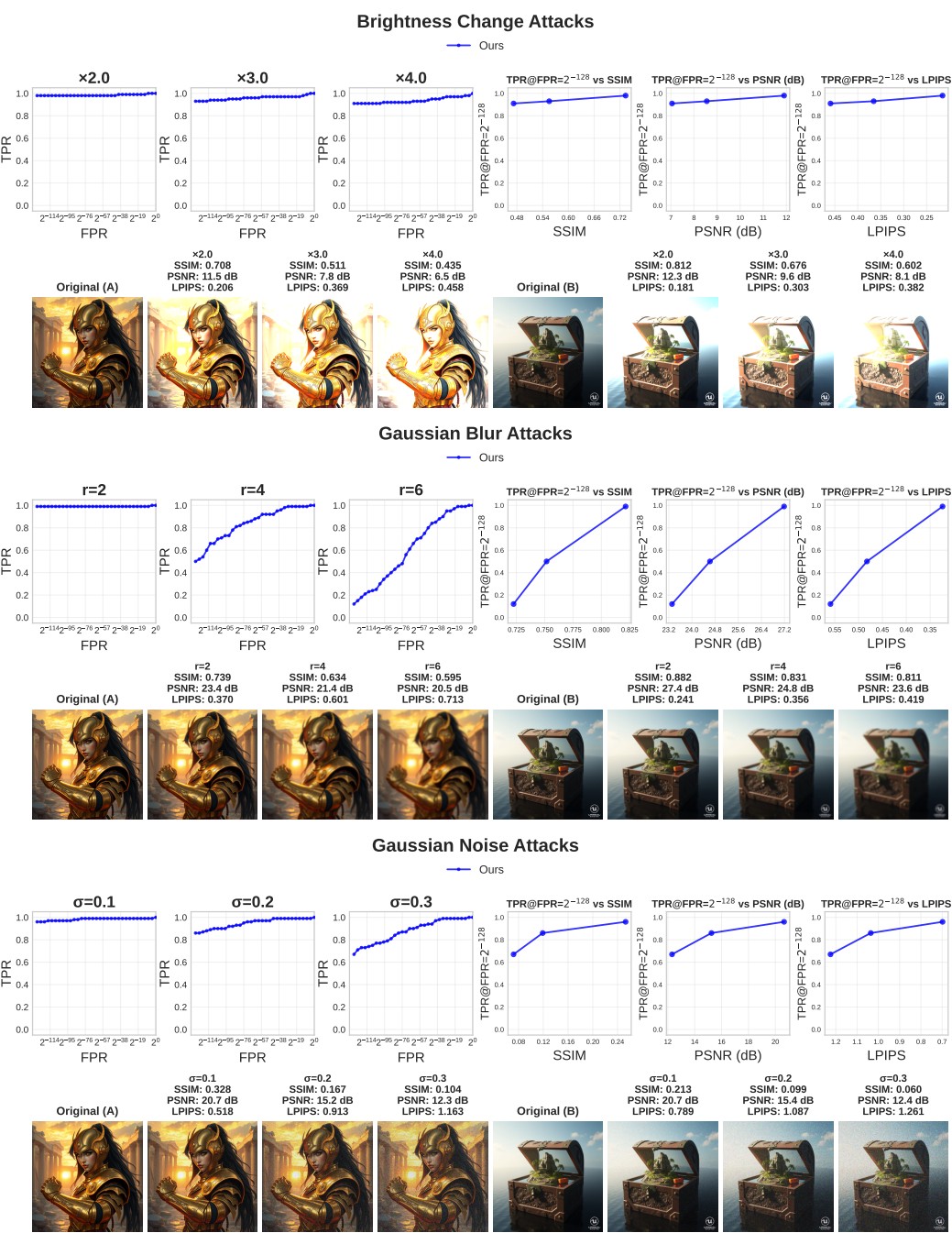

Figure 16: Flux.1-schnell: Evaluating robustness against basic corruption attacks.

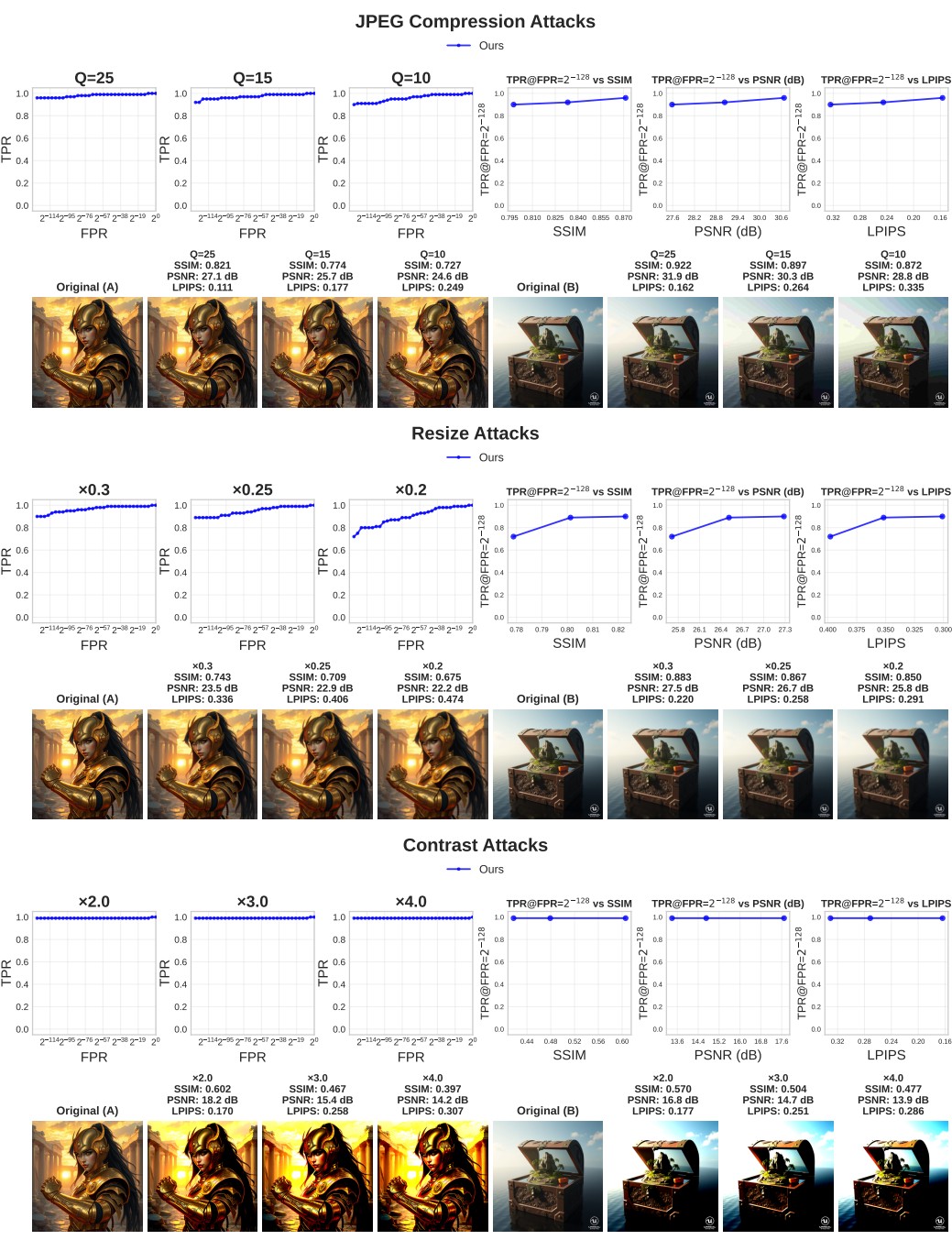

Figure 17: Flux.1-schnell: Evaluating robustness against additional basic corruption attacks.

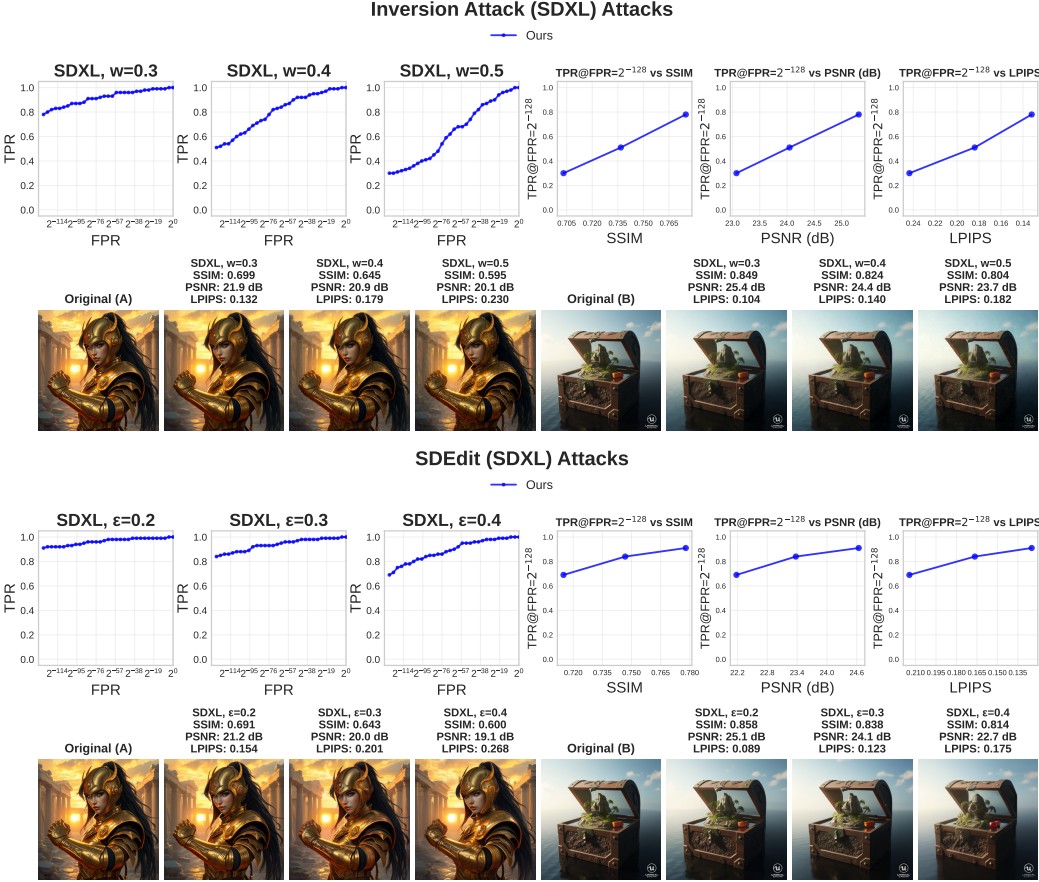

Figure 18: Flux.1-schnell: Evaluating robustness against SDEdit and inversion attacks.

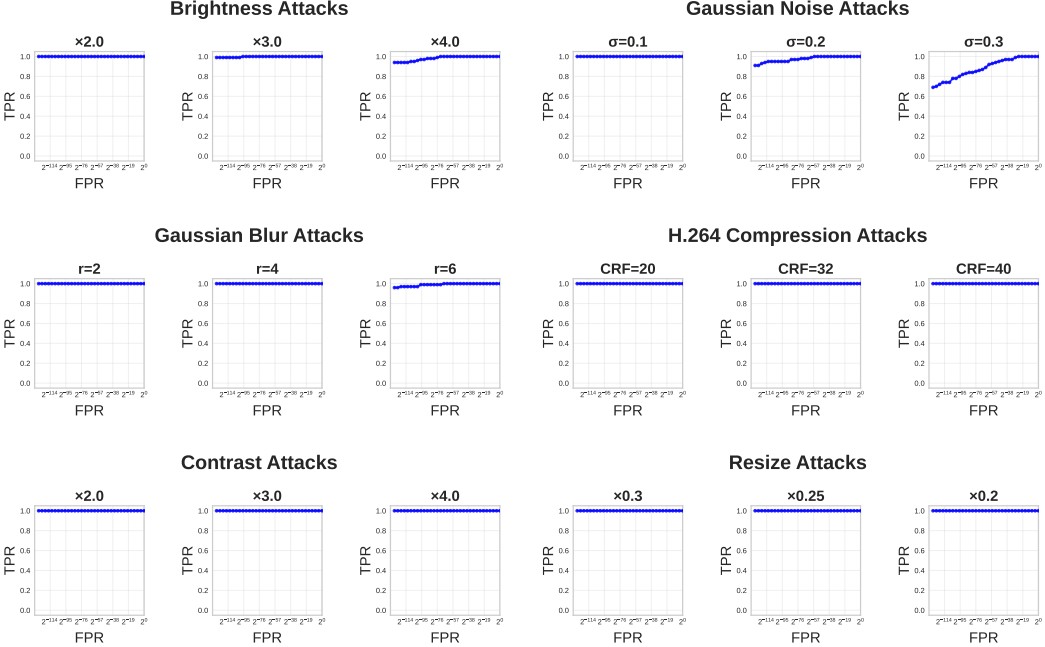

Figure 19: Wan 2.1: Evaluating robustness against basic corruption attacks for video.

## USE OF LLMS

We used ChatGPT to assist with the preparation of this paper. Specifically, we used it to correct grammar and improve sentence-level clarity. We carefully checked all of its outputs. In addition, we used it to help identify related work. All such references were manually verified to exist and were cross-checked against their official sources. All ideas, technical content, and analysis are our own.

