# OpenReview forum: "NoisePrints: Distortion-Free Watermarks for Authorship in Private Diffusion Models"
_ICLR.cc/2026/Conference — ICLR 2026 Poster_

### Official Review · Reviewer_b44h · 2025-10-16

**Soundness:** 3
**Presentation:** 3
**Contribution:** 3
**Rating:** 6
**Confidence:** 3

**Summary:**

This paper provides some insights on how the embedding of the generated image is related to the random seed, and thereby proposes a watermark verification method. The method only verifies the dependency on the embedding and the random seed and no weight information is needed.

**Strengths:**

This paper provides an interesting angle to analyze the relationship between the random seed and the generated image. Theoretical result is also provided for H0 (though I understand there is difficulty in analyzing the underlying distribution under H1).

**Weaknesses:**

(W1) Practicality of the scenario: The paper assumes a setting where the model provider is different from the model user, and the user seeks to protect their IP. However, there are some concerns: (1) If the model provider is untrustworthy, why would the user choose to use that model in the first place? (2) If the model provider is trustworthy, what advantages does this approach offer over existing white-box methods, especially considering potential robustness concerns raised in (W3)?

(W2) Theoretical justification: When $E(x)$ and $h(s)$ are independent, the derived distribution appears reasonable. However, the distribution is unclear when $E(x)$ and $h(s)$ are correlated, leaving a gap in the theoretical analysis.

(W3) Robustness and empirical evaluation: Based on the theoretical results, it seems that the test’s power strongly depends on $E(x)$. This suggests vulnerability to attacks that substantially alter $E(x)$. In the examples in the paper, the embeddings remain relatively close to the originals. For more aggressive attacks, such as redrawing the image, the test’s power could degrade significantly. While the algorithm is lightweight, it would be useful for the authors to provide a comparison that examines the trade-off between computational cost and performance under stronger attacks.

**Questions:**

Please address my concerns in (W1) and (W3). I understand that it would be difficult to derive theories for (W2), but is it possible to provide an empirical distribution (histogram) on the test statistics under H0 and H1 respectively?

---

> ### Author Response · Authors · 2025-11-20
> **Response to Reviewer b44h**
>
> Thank you for your review of our paper. Below we address each of your questions:
>
> ***"Theoretical justification: When E(x) and h(s) are independent, the derived distribution appears reasonable. However, the distribution is unclear when E(x) and h(s) are correlated, leaving a gap in the theoretical analysis."***
>
> Thank you for raising this concern. Please refer to our comment regarding this topic in our general response.
>
> ***"Robustness and empirical evaluation: Based on the theoretical results, it seems that the test's power strongly depends on E(x). This suggests vulnerability to attacks that substantially alter E(x). In the examples in the paper, the embeddings remain relatively close to the originals. For more aggressive attacks, such as redrawing the image, the test's power could degrade significantly. While the algorithm is lightweight, it would be useful for the authors to provide a comparison that examines the trade-off between computational cost and performance under stronger attacks."***
>
> We evaluated our method against a range of common image degradations and adversarial attacks, some of which significantly alter the image content. In the "SD-Edit" attack, the image is partially noised and then denoised using a diffusion model, which can be seen as a form of redrawing. While for the more aggressive configurations of this attack the verification accuracy does degrade, we still observe performance that is better than any of the DDIM-inversion based baselines that we evaluated against, despite our method being much more computationally efficient. In this regard, NoisePrints beats the baselines on both ends of the trade-off spectrum: it is both more robust and more efficient. Of course, we do not claim that our method is robust against all forms of attacks and image manipulations, such as redraw the image completely from scratch while preserving only high-level semantics. We acknowledge in the paper that perfect robustness against adversarial, quality-preserving edits is unattainable [1], but we have emphasized in the revised version of the paper (section 6) that our method is not designed to handle extreme transformations that significantly alter the image content.
>
> ***"is it possible to provide an empirical distribution (histogram) on the test statistics under H0 and H1 respectively?"***
>
> We include empirical distributions (histograms) of the cosine similarity test statistics under both H0 (the cosine similarity between an image and a random noise sample) and H1 (the cosine similarity between an image and the noise sample from which it was generated) in appendix J of the revised paper. This should help illustrate the separation between the two hypotheses and provide a clearer understanding of the verification process. We note that we already provided mean and standard deviation statistics for H1 in Table 1, but histograms will provide a more detailed view of both distributions.
>
> ***"Practicality of the scenario: The paper assumes a setting where the model provider is different from the model user, and the user seeks to protect their IP. However, there are some concerns: (1) If the model provider is untrustworthy, why would the user choose to use that model in the first place? (2) If the model provider is trustworthy, what advantages does this approach offer over existing white-box methods, especially considering potential robustness concerns raised in (W3)?"***
>
> In general, a user may trust the model provider to provide a high-quality generative model, but may still prefer authorship to be verified by a third party. For instance, the user may be concerned with scenarios where the model provider discontinues the model or goes out of business in the future, and thus the user wants to ensure that they can still prove authorship of their generated content without relying on the model provider. As stated in the paper, another scenario is where the model provider does not wish to offer authorship verification services themselves, and instead prefers to delegate this task to a trusted third party verifier. Yet, **our method still provides advantages over existing white-box methods even if the model provider is trusted and willing to provide such services**, since it is significantly more lightweight and efficient when compared to existing methods that require model access during verification, and, as we show in our experiments, is also more robust to some adversarial attacks and image manipulations.
>
> We hope that we have addressed your main concerns. If you feel more positive about our paper, we would appreciate if you would consider updating your score. Otherwise, please let us know if there are any further points you would like us to address or areas for improvement.
>
> [1] Zhang, Hanlin, et al. "Watermarks in the Sand: Impossibility of Strong Watermarking for Generative Models" (2023).

---

> > ### Comment · Reviewer_b44h · 2025-11-22
> >
> > I appreciate the authors in commenting on my questions. I appreciate the authors' contribution in the theoretical perspective, and will maintain my score.

---

### Official Review · Reviewer_j9Rv · 2025-10-29

**Soundness:** 3
**Presentation:** 3
**Contribution:** 2
**Rating:** 2
**Confidence:** 4

**Summary:**

This paper addresses the copyright verification problem for private models and proposes a novel method named NoisePrints. The core idea is to leverage the inherent correlation between the initial Gaussian noise seed used during generation and the final output image, treating this correlation as a natural and verifiable watermark. The authors also innovatively incorporate zero-knowledge proofs, enabling ownership verification in this scenario without exposing the random seed. Experimental results demonstrate that NoisePrints can effectively resist both common image processing operations and specific forgery attack, exhibiting excellent robustness.

**Strengths:**

1. The paper presents a method with an elegantly simple design. Its requirement of only a public VAE for verification makes it highly efficient, resulting in significantly faster performance than existing approaches.
2. The approach eliminates the need for training and preserves output quality, offering a plug-and-play integration capability for diverse diffusion-based architectures.
3. The incorporation of a zero-knowledge proof circuit is a notable strength. It enables secure third-party verification without exposing the secret seed, effectively mitigating a key risk in ownership attestation.

**Weaknesses:**

1. The proposed NoisePrints is a single-bit watermarking method, which is primarily designed for ownership verification. In contrast, many existing methods are multi-bit, offering the extended capability of tracing the specific user responsible for the generation.
2. The proposed scheme faces the same practical deployment challenge as Gaussian Shading[1] (GS). To remain distortion-free, it relies on a randomly sampled seed for each generation, which necessitates securely logging and managing a vast database of `(x, s)` pairs. This requirement creates significant operational overhead and scalability concerns in real-world systems.
3. A logical inconsistency is identified regarding the applicability of the Dispute Protocol. According to Section 3.3, the protocol is activated exclusively when "two parties submit conflicting authorship claims that both pass the verification test." This precondition is inherently incompatible with the threat model of a geometric removal attack, as defined in Section 3.2. In such an attack, the adversary's success is measured by causing the legitimate claim `(x, s)`to fall below the verification threshold τ, thereby making the "both pass" condition unattainable. Consequently, the protocol, as currently formulated, offers no recourse for the rightful owner in this common adversarial scenario.
4. A significant tension exists between the stated threat model and the technical prerequisites of the proposed method. The introduction emphasizes the challenge of watermarking for private models where "model weights remain private and are never shared." However, the verification protocol requires the model provider to use a publicly available VAE encoder. This creates a dependency that contradicts the scenario of a completely self-contained, proprietary model, as a provider wishing to keep their entire pipeline private would be unable to use NoisePrints.
5. A critical security flaw exists in the naive protocol (without ZKP). The requirement for the content producer to expose the seed `s` by submitting the pair `(x, s)` makes the scheme vulnerable to forgery. An adversary who steals this pair can exploit the public VAE to create a perturbed image `x'` that is perceptually similar to `x` but lies outside the verification boundary for the legitimate owner. The adversary can then present the pair `(x', s)` and successfully claim ownership, all without requiring access to the private U-Net. This breaks the security model under a stolen seed scenario.
6. The claimed robustness against geometric attacks is conceptually problematic. The resilience is not an inherent property of the NoisePrint signal but is entirely dependent on the Dispute Protocol's ability to apply a corrective inverse transformation. This process merely reverses a specific, pre-defined manipulation (e.g., rotation, scaling) prior to verification. It does not demonstrate that the watermark itself can survive true geometric distortion, which typically causes irreversible, non-aligned spatial scrambling. The same logic could theoretically be applied to any basic image processing attack (e.g., contrast adjustment, blur) if an effective "inverse operation" could be found and applied. Therefore, the credit lies with the corrective pre-processing within the protocol, not with the fundamental robustness of the NoisePrints method.
7. The evaluation of the Zero-Knowledge Proof (ZKP) implementation is relatively preliminary.
8. The threat model for the "Watermark Injection" attack appears to lack practical motivation. The scenario in which an adversary creates a forged image that is *visually similar* to the original while also embedding an *identical* watermark seems contrived. In practice, an adversary seeking to claim ownership would more plausibly create a *different* image (e.g., a novel artistic creation) and falsely associate it with a forged seed, rather than meticulously replicating the original content with the same watermark. The authors should either provide a stronger justification for the considered injection attack scenario or redefine it to reflect a more realistic adversarial goal.

**Questions:**

1. **Regarding Weakness 1**: NoisePrints already assigns a unique seed to each user as an identity identifier. This foundation could be directly extended to construct a simple multi-bit scheme, for instance, by allocating a subset of seeds to represent specific user IDs. However, the authors have not evaluated NoisePrints from this perspective. To comprehensively benchmark against state-of-the-art methods like Stable Signature[2] and Gaussian Shading[1], it is necessary to evaluate its performance in terms of traceability accuracy within a multi-bit framework.
2. **Regarding Weakness 3**: The description of the Dispute Protocol's usage scenario should be reorganized to align with the experimental setup and resolve the logical contradiction when facing geometric attacks.
3. **Regarding Weakness 4**:  To resolve the contradiction in the threat model, I recommend removing the requirement for a public VAE. Instead of relying on model providers to reuse a public VAE—which conflicts with the scenario of fully proprietary models—the model owner could entrust their private VAE to the fully trusted verifier. This approach would better align with the stated principle that "the verifier is the only trusted party" and that "model weights remain private and are never shared," while still enabling the verification procedure.
4. **Regarding Weakness 5**:  I recommend that the authors augment the threat model with specific strategies to mitigate the risk of seed exposure in the naive protocol version.
5. **Regarding Weakness 6:** The paper attributes geometric robustness to the NoisePrints method itself. However, the described mechanism relies entirely on the Dispute Protocol's ability to apply an inverse transformation to "correct" the image before verification. Could you clarify how this approach demonstrates inherent robustness of the watermark signal, as opposed to being a general pre-processing strategy that could theoretically be applied to any watermarking scheme? Furthermore, does this mean that NoisePrints' geometric robustness is ultimately limited to attacks that are both invertible and whose inverse transformation is known and included in the public set 𝒢?
6. **Regarding Weakness 7:** The paper demonstrates the functional correctness of the ZKP implementation. However, its security guarantees are primarily cryptographic. A critical remaining question is: how does the *watermark robustness* fare when the image undergoes attacks *before* the ZKP-based verification is performed? Specifically, if an attacked image `x'` (e.g., after JPEG compression, blurring, or a geometric transformation) is submitted for ZKP verification, will the circuit still correctly output `1` (indicating a valid watermark) when provided with the legitimate seed `s`? We recommend that the authors conduct a simple but essential robustness evaluation for the ZKP scenario to confirm that the strong robustness demonstrated in the standard setting is preserved when verification occurs within the ZKP circuit.
7. **Regarding Weakness 8:** The authors should consider evaluating more practical forgery attacks, such as those proposed in [3].

8.  The paper does not mention a specific method for mapping bits $PRNG(h(s))$ to Gaussian noise $\epsilon(h(s))$. How is this step concretely implemented?
9. It is unclear from the experimental description whether a single VAE was used for all verification tasks, or whether the native VAE corresponding to each generative model was employed. Could the authors clarify this point?

[1] Yang, Zijin, et al. "Gaussian shading: Provable performance-lossless image watermarking for diffusion models." *Proceedings of the IEEE/CVF Conference on Computer Vision and Pattern Recognition*. 2024.

[2] Fernandez, Pierre, et al. "The stable signature: Rooting watermarks in latent diffusion models." *Proceedings of the IEEE/CVF International Conference on Computer Vision*. 2023.

[3] Müller, Andreas, et al. "Black-box forgery attacks on semantic watermarks for diffusion models." *Proceedings of the Computer Vision and Pattern Recognition Conference*. 2025.

---

> ### Author Response · Authors · 2025-11-20
> **Response to Reviewer j9Rv [1/3]**
>
> Thank you for your review of our paper. Below we address each of your questions:
>
> ***"NoisePrints already assigns a unique seed to each user as an identity identifier. This foundation could be directly extended to construct a simple multi-bit scheme, for instance, by allocating a subset of seeds to represent specific user IDs. However, the authors have not evaluated NoisePrints from this perspective. To comprehensively benchmark against state-of-the-art methods like Stable Signature and Gaussian Shading, it is necessary to evaluate its performance in terms of traceability accuracy within a multi-bit framework."***
>
> In our paper, we focused on allowing users to prove authorship of generated content in a model-free manner. However, We agree that extending NoisePrints to a multi-bit watermarking scheme is straightforward. We have added an experiment showcasing how NoisePrints can be used for tracing which user generated a specific image in a multi-user setting in the revised paper (appendix I), demonstrating its strong performance in this setting.
>
> ***"The description of the Dispute Protocol's usage scenario should be reorganized to align with the experimental setup and resolve the logical contradiction when facing geometric attacks."***
>
> We have reorganized the description of the Dispute Protocol to better align with the experimental setup and clarify how it addresses geometric attacks. Specifically, when and how the Dispute Protocol is invoked in the presence of such attacks is now more clearly outlined to avoid any confusion. Furthermore, we revised the description of how geometric transforms are dealt with within the protocol, which is now better aligned with the experimental setup.
>
> "***To resolve the contradiction in the threat model, I recommend removing the requirement for a public VAE. Instead of relying on model providers to reuse a public VAE—which conflicts with the scenario of fully proprietary models—the model owner could entrust their private VAE to the fully trusted verifier. This approach would better align with the stated principle that "the verifier is the only trusted party" and that "model weights remain private and are never shared," while still enabling the verification procedure."***
>
> Thank you for this valuable suggestion, please refer to our comment regarding this topic in the global response.
>
> ***"The threat model for the "Watermark Injection" attack appears to lack practical motivation. The scenario in which an adversary creates a forged image that is visually similar to the original while also embedding an identical watermark seems contrived. In practice, an adversary seeking to claim ownership would more plausibly create a different image (e.g., a novel artistic creation) and falsely associate it with a forged seed, rather than meticulously replicating the original content with the same watermark. The authors should either provide a stronger justification for the considered injection attack scenario or redefine it to reflect a more realistic adversarial goal."***
>
> We appreciate the reviewer's comment and recognize that our description of the "Watermark Injection" attack may have been unclear. We would like to clarify the threat model we consider:
> The adversary does not embed an identical watermark to the original. Rather, the attack scenario involves an adversary who:
>
> 1. Obtains an image created and published by a legitimate user (e.g., posted on their website)
>
> 2. Creates a visually similar forged image
>
> 3. Embeds a new watermark associated with a different forged seed $s'$ (not the original seed $s$)
>
> 4. Falsely claims authorship of this forged image (e.g., to use it commercially without compensating the original creator)
>
> This attack is practically motivated: the adversary seeks to profit from content that closely resembles the original work by injecting their own watermark into a modified version. The dispute protocol addresses precisely this scenario: an adversary claims ownership over modified content and passes the basic verification protocol by virtue of having injected their own watermark.
>
> We hope that together with the reorganization of the dispute protocol description in the paper, this helps to clear up our intention regarding the threat model presented in our paper.

---

> ### Author Response · Authors · 2025-11-20
> **Response to Reviewer j9Rv [2/3]**
>
> ***"I recommend that the authors augment the threat model with specific strategies to mitigate the risk of seed exposure in the naive protocol version."***
>
> We recognize that seed exposure is a critical concern in the naive protocol version. Exactly due to this concern, in scenarios where the seed being exposed is a significant risk, we propose using the ZKP-based verification protocol. The ZKP-based protocol is introduced in the paper after the naive protocol in order to not burden readers with the additional complexity of ZKPs before they understand the basic idea of NoisePrints. However, in the threat model section of the revised paper we have now emphasized the importance of avoiding seed leakage, and that using our proposed ZKP-based protocol is the recommended approach to reduce the chance of the seed leaking.
>
> ***"The paper attributes geometric robustness to the NoisePrints method itself. However, the described mechanism relies entirely on the Dispute Protocol's ability to apply an inverse transformation to "correct" the image before verification. Could you clarify how this approach demonstrates inherent robustness of the watermark signal, as opposed to being a general pre-processing strategy that could theoretically be applied to any watermarking scheme? Furthermore, does this mean that NoisePrints' geometric robustness is ultimately limited to attacks that are both invertible and whose inverse transformation is known and included in the public set G?"***
>
> While we agree with the reviewer that this approach could theoretically be applied to other watermarking scheme in the discussed setting, we consider the proposed mechanism to be a part of our complete proposed solution for allowing users to prove authorship over images that underwent geometric alterations. Indeed, dealing with a geometric transformation via the dispute protocol relies on the ability to apply an inverse transformation to "align" the image before verification, using some inverse transformation from the public set G (as we acknowledge in the paper). In Appendix H, we demonstrate that this approach is effective in practice for rotation and crop & scale transformations, and the inverse transformation can easily be obtained with access to the reference image (which is applicable in the setting presented in the paper). It should be noted that other distortion-free watermarking schemes also struggle with geometric transformations, and are often evaluated under rotation or sliding window searches (see e.g Appendix H in [1]), or assuming alignment with the original image (see e.g Random Crop evaluation in [2]).
>
> [1] Arabi, Kasra, et al. "Hidden in the Noise: Two-Stage Robust Watermarking for Images" (2024).
>
> [2] Yang, Zijin, et al. "Gaussian Shading: Provable Performance-Lossless Image Watermarking for Diffusion Models" (2024).

---

> ### Author Response · Authors · 2025-11-20
> **Response to Reviewer j9Rv [3/3]**
>
> ***"The authors should consider evaluating more practical forgery attacks, such as those proposed in [3]."***
> Response: We evaluated our method against commonly used image manipulations, as well as more sophisticated adversarial attacks. The removal method proposed in [3] is actually very similar in its approach to the DDIM-inversion based adversary attack that we proposed and evaluated against in our paper. In both methods, an approximation of the initial latent is obtained with DDIM inversion using a proxy model. In [3], the image latents are optimized using gradient descent to minimize the correlation of the inverted noise latents with the initial noise approximation. In our proposed attack, the image latents are optimized using gradient descent to minimize their direct correlation with the initial noise approximation - which aims to directly make our proposed verification fail. Yet surprisingly, our method displays greater resilience to this attack when compared to competing DDIM inversion based methods. For completeness, we will also add an evaluation against the adversarial attack proposed in [3]. As this attack requires a lengthy optimization process per attacked image and given limited compute resources, it will take us a few days to obtain the results.
>
> ***"The paper does not mention a specific method for mapping bits PRNG(h(s)) to Gaussian noise ε(h(s)). How is this step concretely implemented?"***
>
> Most of our experiments use the PyTorch implementation of torch.randn accepting a torch.Generator initialized with a random seed. torch.randn internally uses the Box-Muller method. For the ZKP experiments, our mapping from bits to Gaussian noise is based on [4] as we mentioned in Appendix C.1. We have included those details in appendix K in the revised version of the paper.
>
> ***"It is unclear from the experimental description whether a single VAE was used for all verification tasks, or whether the native VAE corresponding to each generative model was employed. Could the authors clarify this point?"***
>
> In all our experiments, we used the native VAE corresponding to each generative model for verification tasks.
>
> We hope that we have addressed your main concerns. If you feel more positive about our paper, we would appreciate if you would consider updating your score. Otherwise, please let us know if there are any further points you would like us to address or areas for improvement.
>
> [3] Müller, Andreas, et al. "Black-Box Forgery Attacks on Semantic Watermarks for Diffusion Models" (2024).
>
> [4] Acklam, Peter, "An algorithm for computing the inverse normal cumulative distribution function." (2003).

---

> ### Author Response · Authors · 2025-11-26
> **Evaluation Results for "Black-Box Forgery Attacks on Semantic Watermarks for Diffusion Models" Imprint-Removal Attack**
>
> Dear reviewer,
>
> We would like to inform you that we have added the evaluation results on the imprint-removal attack proposed in [1] as we committed to do in our previous response. We have added the evaluation results under Appendix L of the revised paper (Figure 15). As can be seen, our method performs significantly better against this adversarial attack when compared to competing inversion based methods (against which the attack is highly effective). Our method is able to retain high pass rates after the attack is applied, even at a threshold corresponding to extremely low FPR of $2^{-128}$.
>
> We hope we were able to address your concerns, and welcome further discussion if you have any other questions or if there are additional points you would like us to address.
>
> [1] Müller, Andreas, et al. "Black-Box Forgery Attacks on Semantic Watermarks for Diffusion Models" (2024).

---

### Official Review · Reviewer_8Bov · 2025-10-30

**Soundness:** 2
**Presentation:** 2
**Contribution:** 2
**Rating:** 6
**Confidence:** 3

**Summary:**

The paper tackles the challenge of proving authorship over visual AI-generated content, especially for cases where the diffusion model is private and traditional watermarking approaches (which need model weights or modify outputs) are impractical or inefficient. The authors propose NoisePrints, a lightweight, distortion-free watermarking scheme that utilizes the random seed used to initialize the diffusion process as a watermark proof, exploiting the strong correlation between this seed’s noise and the generated content. The verification only requires the seed and output, with no changes to the generation process and no access to model internals. NoisePrints is validated on multiple state-of-the-art diffusion models for images and videos, demonstrating efficient verification using only the seed and output, without requiring access to model weights.

**Strengths:**

**Originality:**

NoisePrints introduces a technically novel approach by using the stochastic seed of diffusion models as a watermark, enabling efficient and model-agnostic authorship verification. The scheme also integrates cryptographic techniques, such as zero-knowledge proofs, to assure privacy and security in third-party verification scenarios.

**Quality:**

The paper demonstrates strong methodological rigor, offering a comprehensive security analysis and empirical validation across diverse models and datasets. Experiments show the watermark’s high robustness to output manipulations and adversarial attacks, while significantly reducing computational overhead compared to inversion-based methods.

**Clarity:**

The writing is clear and logical. Core ideas, threat models, and algorithms are well explained, with protocols articulated stepwise, making the contributions accessible to both technical and broader audiences.

**Significance:**

By removing the need for model internals and ensuring distortion-free watermarking, NoisePrints directly addresses real-world needs for copyright and provenance management in private and proprietary diffusion models. Its privacy-preserving design and scalability make it highly significant for the trustworthy deployment of generative AI and digital content protection.

**Weaknesses:**

- In the introduction (from line 52), the discussion of the method does not clearly convey the technical challenges involved. As a result, readers may be left with the impression that the approach is straightforward to implement, which could undermine the perceived significance of the contribution. The authors should better articulate the complexities and nontrivial aspects of their method.
- Tables 1 and 2 lack visual highlights or markers for the best-performing methods, making it difficult for readers to quickly identify key results. Clear visual cues, such as bolding or color highlights, are recommended to enhance table readability and emphasize the main findings.
- Sections 3.3 and 3.4 are dense with technical details and formalism, posing accessibility challenges for readers without deep expertise in diffusion models or cryptography. These sections would benefit from additional intuitive diagrams and plain-language protocol summaries to broaden accessibility.
- Figure 1 suffers from inconsistent font usage and the inclusion of citation notes within module labels, which detracts from the professionalism and clarity of the visual. The authors should standardize fonts and reconsider the figure layout for improved visual coherence.
- Figures 2 and 3 are overly large and contain dense information, resulting in plot axes that are difficult to read. The authors should revise these figures to more logically group and present the data, possibly by dividing them into multiple panels and increasing the size and clarity of key elements.

**Questions:**

1. As mentioned in the limitation: The verification scheme relies on access to the public VAE used by the diffusion model. When the VAE is not public or is heavily modified, the approach may be less applicable. How the author plans to address such issues?
2. Can the authors better articulate the complexities and nontrivial aspects of their method in terms of the task of this paper.

---

> ### Author Response · Authors · 2025-11-20
> **Response to Reviewer 8Bov**
>
> Thank you for your review of our paper. We have improved the presentation of our paper following your comments, and will continue to further touch-up the presentation towards the camera-ready version of the paper.
> Below we address each of your questions:
>
> ***"As mentioned in the limitation: The verification scheme relies on access to the public VAE used by the diffusion model. When the VAE is not public or is heavily modified, the approach may be less applicable. How the author plans to address such issues?"***
>
> We thank you for raising this concern. Please refer to our comment regarding this topic in our global response.
>
> ***"Can the authors better articulate the complexities and nontrivial aspects of their method in terms of the task of this paper."***
>
> At its core, our method is concise and easy to reproduce. We view this as a strength of our approach, as it allows for easy adoption and integration into existing diffusion model pipelines. Our contribution lies in the novel conceptualization of leveraging the inherent correlation between the initial noise latents and the generated content latents for model-free, lightweight and distortion free watermarking purposes, designing a dispute resolving protocol that handles injection attempts and geometric transforms, as well as incorporating zero-knowledge proofs as part of the verification for additional security and the implementation thereof. We have revised the section in the introduction where we first present NoisePrints to mention all those aspects.
>
> We hope that we have addressed your main concerns. If you feel more positive about our paper, we would appreciate if you would consider updating your score. Otherwise, please let us know if there are any further points you would like us to address or areas for improvement.

---

> > ### Comment · Reviewer_8Bov · 2025-11-26
> >
> > Thanks to the authors for addressing my concerns! I would like to keep my original rating.

---

### Official Review · Reviewer_cyNm · 2025-10-31

**Soundness:** 2
**Presentation:** 3
**Contribution:** 2
**Rating:** 4
**Confidence:** 3

**Summary:**

This paper proposes NoisePrints, watermarking framework for diffusion models that uses the random seed of the generation process as a proof of authorship. By leveraging the strong correlation between the initial noise and the final output, the proposed method secures the generation process with a cryptographic hash and optional zero-knowledge proof, which enables verification without accessing diffusion model weights. Experiments on image and video diffusion models demonstrate that NoisePrints achieves robust and efficient authorship verification under common content manipulations.

**Strengths:**

1. The writing of the paper is clear and well-structured.
2. The proposed method is simple but effective, especially in terms of robustness against different types of attacking.
3. Experiments are conducted on multiple diffusion models (including both image and video generation) and against various types of attacks, demonstrating the generality of the method.

**Weaknesses:**

1. A main concern is the applicability of the method, in the scenario considered in this paper, the verification of the watermark relies on public structure VAE. However, in practice, it is possible that some diffusion models may update or fine-tune their VAEs across versions. It is not clear that under this circumstance, whether the proposed method is still effective.
2. The motivation for the considered scenario requires stronger justification. In real-world applications, a more common concern is that model owners aim to trace who is responsible for the malicious or unauthorized use of their models, or that data owners wish to verify whether their data have been improperly used to train a model. In contrast, if a regular user simply generates an image using the model, it is unclear why they would need to prove authorship of the generated content, or why others might contest such authorship.

**Questions:**

1. If the watermarked images gone through semantic level modification, such as style transfer, can the watermarked detection accuracy still maintain?
2. According to some studies [1], large diffusion models exhibit partial memorization of training images. In such cases, different seeds may yield visually or latently similar outputs, breaking the one-to-one correspondence between the seed and the generated content. This could lead to higher false positive rates in NoisePrints verification, since unrelated seeds might still produce embeddings that correlate above the verification threshold. Can the author provide evaluation results of proposed method in such case?

[1] Memory triggers: Unveiling memorization in text-to-image generative models through word-level duplication
[2] Understanding (un) intended memorization in text-to-image generative models.
[3] Extracting training data from diffusion models.

---

> ### Author Response · Authors · 2025-11-20
> **Response to Reviewer cyNm**
>
> Thank you for your review of our paper. Below we address each of your questions:
>
> ***"A main concern is the applicability of the method, in the scenario considered in this paper, the verification of the watermark relies on public structure VAE. However, in practice, it is possible that some diffusion models may update or fine-tune their VAEs across versions."***
>
> We thank you for raising this concern. Please refer to our comment regarding this topic in our global response.
>
> ***"The motivation for the considered scenario requires stronger justification. In real-world applications, a more common concern is that model owners aim to trace who is responsible for the malicious or unauthorized use of their models, or that data owners wish to verify whether their data have been improperly used to train a model. In contrast, if a regular user simply generates an image using the model, it is unclear why they would need to prove authorship of the generated content, or why others might contest such authorship."***
>
> The motivations for watermarking generative models are diverse and multifaceted. In our paper, we focused on a specific scenario where users may wish to assert authorship over generated content, particularly in contexts where content authenticity is critical (e.g., digital art, media production). However, we acknowledge that other scenarios, such as tracing unauthorized use or verifying data provenance, are equally important. Thankfully, our proposed watermarking scheme is versatile and can be adapted to address these concerns as well, as we mention in the Conclusion section. We have added an experiment showcasing how NoisePrints can be used for tracing which user generated a specific image in a multi-user setting in the revised paper, demonstrating its strong performance in this setting.
>
> ***"If the watermarked images gone through semantic level modification, such as style transfer, can the watermarked detection accuracy still maintain?"***
>
> While our current experiments primarily focus on perturbations that retain structural similarity with the original content, we acknowledge that semantic-level modifications, such as style transfer, present a more challenging scenario for watermark detection. Such transformations may be considered as artistic derivations rather than adversarial attacks, and our method is not well suited to handle them at present. However, it should be noted that methods that rely on watermarks embedded in the initial noise latents are also likely to struggle with such modifications, and as shown in previous work [1], no watermarking method can be robust against an unrestricted adversary. We discuss this limitation in the revised version of the paper (section 6).
>
> ***"According to some studies, large diffusion models exhibit partial memorization of training images. In such cases, different seeds may yield visually or latently similar outputs, breaking the one-to-one correspondence between the seed and the generated content. This could lead to higher false positive rates in NoisePrints verification, since unrelated seeds might still produce embeddings that correlate above the verification threshold. Can the author provide evaluation results of proposed method in such case?"***
>
> Our FPR analysis holds regardless of the mapping between noise and output of the model. Please refer to our comment regarding this topic in the general response.
>
> We hope that we have addressed your main concerns. If you feel more positive about our paper, we would appreciate if you would consider updating your score. Otherwise, please let us know if there are any further points you would like us to address or areas for improvement.
>
> [1] Zhang, Hanlin, et al. "Watermarks in the Sand: Impossibility of Strong Watermarking for Generative Models" (2023).

---

### Author Response · Authors · 2025-11-20
**Global Response**

We thank all the reviewers for their valuable feedback. Our work introduces NoisePrints, a method that takes advantage of the inherent correlation between the initial noise and the generated content in order to provide robust, efficient, distortion-free watermarking verifiable by a third party without access to the denoising model. We are glad that the reviewers found our method elegant (j9Rv) effective and robust (cyNm,8Bov,j9Rv), and found our paper clear (cyNm,8Bov) and interesting (b44h).

Based on your feedback, we have uploaded a revised version of the paper. **For your convenience, new and modified sections in the paper are highlighted in blue text.**

Below we address two common topics that were raised by the reviewers:

(A) **Requirement of a public VAE for verification**

Some reviewers raised concerns regarding the requirement that the VAE is public, which may limit the applicability of our method in certain scenarios.

While in the paper we framed having access to a public VAE as a requirement, following j9Rv’s suggestion, we want to clarify that in practice **the VAE model weights need not be public**. Instead, it suffices that the verifying party is able to encode the image through the model's VAE, potentially via restricted API access, in order to perform the cosine similarity check against the noise latents produced by the provided seed. This formulation of the requirement is more minimal, precise, and realistic, and we have adjusted it in the revised version.

(B) **False positive rate analysis**

Some reviewers were concerned about the correctness of our false positive rate (FPR) analysis, and were under the impression that it only holds assuming that the noise distribution and the distribution of generated images are independent, an assumption which does not hold in practice. However, **we do not make such an assumption in our analysis**.

Our theoretical analysis concerns only the false positive rate of the verification scheme given a fixed threshold. That is, the probability for an arbitrary image x (regardless of how it was generated) to be incorrectly verified as being generated from a randomly drawn seed $s$. Our proof in Appendix B shows that the FPR bound holds for **any arbitrary vector** in the latent space, and only assumes that the noise distribution is that of an i.i.d Gaussian. For the sake of completeness, we also added in appendix J histograms of the cosine similarity between image latents and a) their original noise sample and b) unrelated random noise samples.

We further address the individual concerns raised by each reviewer in a response to their review.

---

### Comment · Area_Chair_kSam · 2025-11-25

Dear Reviewers,

The authors have submitted their responses to your questions and feedbacks. Please read them and give your comments.

Regards, AC

---

### Meta-Review · Area_Chair_6N4c · 2026-01-06

**Summary:**

The paper proposes a light-weight watermark method for diffusion models. The paper presents an interesting solution to an important problem in watermarking diffusion models without training. The theoretical analysis of zero-knowledge also makes the proposed method more reliable. The reviewers showed some concerns about the impractical setting where the generation needs to be done in a publicly available VAE structure. Also, there are some concerns about the watermark robustness analysis, where only some image editing methods have been tested. The reviewers also mentioned that the multi-bit scenario needs to be added for a more comprehensive comparison. The rebuttal solves a lot of the aforementioned concerns, and I believe the paper could be a good addition if the comments are carefully incorporated into the revision.

**Reviewer Concerns:**

The authors did a good job in adding several adversarial attacks on the robustness analysis and added a preliminary experiment on the multi-bit framework, which helps to address the concerns and needs to be put in the final revision. However, the concern about the impractical setting where the generation needs to be done in a publicly available VAE structure still exists.

**Reviewer Scores:**

Two positive reviewers have already kept their score, and I expect the Reviewer j9Rv will increase to score to at least 4. I believe Reviewer cyNm will keep the score since the main concern about the setting is not addressed yet.

---

### Decision · Program_Chairs · 2026-01-26

Accept (Poster)